# Research Progress on Bacteria-Reducing Pretreatment Technology of Meat

**DOI:** 10.3390/foods13152361

**Published:** 2024-07-26

**Authors:** Hong Zuo, Bo Wang, Jiamin Zhang, Zhengguo Zhong, Zhonghua Tang

**Affiliations:** 1Meat Processing Key Laboratory of Sichuan Province, Chengdu University, Chengdu 610106, China; zuohong@stu.cdu.edu.cn (H.Z.); wangbo9214@163.com (B.W.); 2BaHang Food Development Co., Zigong 643030, China; 3SiChuan AiChiTu Food Co., Bazhong 636600, China; 13629028888@163.com

**Keywords:** meat storage and preservation, non-thermal sterilization technology, sterilization and preservation, bacteria-reducing technology

## Abstract

Reducing the initial bacteria number from meat and extending its shelf life are crucial factors for ensuring product safety and enhancing economic benefits for enterprises. Currently, controlling enzyme activity and the microbial survival environment is a common approach to reducing the rate of deterioration in raw meat materials, thereby achieving the goal of bacteria reduction during storage and preservation. This review summarizes the commonly used technologies for reducing bacteria in meat, including slightly acidic electrolyzed water (SAEW), organic acids, ozone (O_3_), ultrasound, irradiation, ultraviolet (UV), cold plasma, high-pressure processing (HPP), and biological bacterial reduction agents. This review outlines the mechanisms and main features of these technologies for reducing bacteria in meat processing. Additionally, it discusses the status of these technologies in meat storage and preservation applications while analyzing associated problems and proposing solutions. The aim is to provide valuable references for research on meat preservation technology.

## 1. Introduction

In recent years, there has been a rapid increase in demand for various types of meat and meat products. Meat contains a large amount of fat, protein, vitamins, minerals, and other nutrients [1]. These nutrients can be utilized by foodborne microbes such as *Pseudomonas*, *Acinetobacter*, *Enterobacter*, *Lactobacillus*, and other microorganisms to grow and reproduce, which not only causes the decrease in raw meat character but also shortens the shelf life of the meat [2]. The shelf life of meat is closely related to the initial number of microorganisms before storage. The effective sterilization pretreatment technology of meat raw materials and meat products without high-temperature sterilization can reduce the initial amount of bacteria in meat raw materials, which is an effective means to inhibit the growth of microorganisms in meat and prolong its shelf life.

At present, the common bactericidal technologies at home and abroad mainly include chemical bacteria reduction technology (slightly acidic electrolyzed water, organic acids, ozone), non-thermal physical bacteria reduction technology (ultrasound, irradiation, ultraviolet, cold plasma, high-pressure processing), and biological bacteria reduction technology (plant-derived natural antibacterial agents, animal-derived natural antibacterial agents, microbial source natural antibacterial agents, etc. [3]). These technologies for reducing bacteria can temporarily or permanently disrupt the living conditions of microorganisms by altering the internal state or external environment of meat. They not only effectively reduce the number of microorganisms in meat but also ensure better food quality. However, the application of a single bacterial reduction technology may reduce the antibacterial effect due to the induced stress response of bacteria, while the combined application of multiple bacteria-reducing technologies can avoid the above drawbacks to a certain extent. The hurdle technology is a method for ensuring the safety of food products by controlling or destroying microbial spoilage or food pathogens through the combination of a series of preservative parameters (hurdles) imposing physiological impact on microbial cells [4,5]. Many researchers utilize physical sterilization technology, chemical bactericidal technology, biological bactericidal technology, and low-temperature preservation technology as a combination of defense factors to achieve the preservation effect of “1 + 1 > 2” through fence technology [6,7,8,9]. In the process of meat storage, it is necessary to scientifically select the appropriate bacteria-reducing technology based on the type of meat, processing methods, and storage conditions.

Databases, such as Web of Science, Scopus, PubMed, and Science Direct, were used to systematically search the literature. The search methods included keyword search (bacteria reduction technology, bacteria/non-thermal physical/reduction technology, organic acid, ozone, ultrasonic, etc.), journal search (foods, food chemistry, meat science, etc.), and citation literature search (based on the situation of the target article cited by other articles). Previous studies have extensively examined the application of single bacteriological reduction technologies in food [10,11,12]. However, this paper reviews the principles, mechanisms, and applications of various bacteriological reduction technologies in meat preservation. Furthermore, it discusses the advantages and disadvantages of these bacteria-reducing technologies as well as their future development trends. The aim is to provide a scientific theoretical reference for better understanding and application of bacteria-reducing technologies in meat preservation.

## 2. Classification of Bacteria-Reducing Technologies and Their Mechanisms for Reducing Bacteria

### 2.1. Chemical Bacteria Reduction Technology

Chemical bacteria reduction technology refers to the use of biochemical agents to kill or inhibit the growth and reproduction of microorganisms in the environment, on the surface or inside the raw materials, so that the physicochemical reactions and metabolic activities related to microbial cells are interfered with and destroyed, which ultimately leads to the death of microorganisms. The application in the field of food is mainly used directly for food or disinfection of the food production process, which has become a key factor in reducing bacteria in food raw materials. Currently, commonly used chemical bacteriological reduction technologies for meat raw materials include slightly acidic electrolyzed water, organic acids, ozone, and so on.

Slightly acidic electrolyzed water is electrolyzed water with a certain available chlorine content (ACC) by applying a direct current voltage to an electrolytic cell without a diaphragm [13]. Its pH is generally 5.0~6.5, and its oxidation–reduction potential (ORP) ≥ 600 mV. It has been found that the HCIO in SAEW, which exhibits a highly potent bactericidal effect, can result in a microbial lethality rate of over 90% [14]. SAEW attacks the cell wall, cell membrane, and intracellular components of microbial cells to achieve the purpose of sterilization, effectively reducing the common foodborne pathogens in food. Moreover, due to the short action time of SAEW, it will not cause bacterial resistance to develop. Therefore, it is a chemical technology for reducing bacteria with broad prospects for development. However, strict control of the pH range of SAEW is necessary to avoid compromising its bactericidal effects. Additionally, long-term use may result in corrosion and damage to metal equipment surfaces.

Organic acids, as food grade antibacterial agents, are a class of organic compounds that contain carboxylic (-COOH) functional groups. They can inhibit the growth of microorganisms by reducing the pH value of food and play a bactericidal role. Organic acids commonly used for reducing bacteria in food include lactic acid, propionic acid, sorbic acid, acetic acid, citric acid, and tartaric acid. These acids are primarily produced through microbial fermentation engineering or naturally occurring processes. They are widely employed in food preservation and are internationally recognized as safe (GRAS) bactericidal agents [15]. In addition to their role as bacterial-reducing agents in food, organic acids are also listed in Food and Drug Administration (FDA) regulations for various technical applications, such as acidity regulators, antioxidants, flavor enhancers, and pH adjusters [16]. It should be noted that relatively high concentrations of certain organic acids may affect the taste and flavor of food, as well as potentially persist in it, thus requiring reasonable control during usage.

Ozone is a light blue gas with a distinct odor, which is produced by an ozone generator through high voltage discharge or ultraviolet radiation alone or in combination with nitrogen oxides (NOx) [17]. Ozone is commonly used in both gaseous and aqueous forms in the food industry. It acts on cell walls, cell membranes, intracellular enzymes, genetic material, fungal spore shells, and viral capsids to inactivate a variety of microorganisms through ozone itself or its breakdown products, such as hydroxyl radicals [18]. It is also easy to decompose into HO- and O_2_ in water and has a strong disinfecting and sterilizing ability [19]. Ozone automatically and rapidly breaks down in the air to produce oxygen, leaving no residue in the food. Furthermore, ozone can quickly diffuse throughout space, ensuring sterilization without any missed areas. Due to its high bactericidal efficiency [20], ozone is widely utilized in food preservation and other fields. However, ozone treatment requires specialized equipment and technology. The operation process is relatively complex, and high concentrations of ozone are harmful to the human body. Therefore, strict control of the concentration and contact time is necessary when using it.

### 2.2. Non-Thermal Physical Bacteria Reduction Technology

Non-thermal physical sterilization technology does not require the addition of chemical substances, which can effectively prevent human health hazards caused by chemical residues. Compared with traditional thermal sterilization technology, usually at low temperature or room temperature, it can achieve the purpose of sterilization and can better ensure the quality of food. Commonly used methods for sterilizing meat raw materials include ultrasound, radiation, ultraviolet light, cold plasma, high-pressure processing, etc.

Ultrasound is a longitudinal mechanical wave beyond the range of human hearing, with a frequency generally between 20 kHz and 1 GHz [21]. It can travel through air, liquids, and solids and has the capability to kill certain microorganisms in food. In the food industry, ultrasound at a lower frequency (20~100 kHz) is usually used to deactivate microorganisms. The bactericidal mechanism of ultrasonic waves is mainly attributed to the cavitation effect [22]. Under the influence of ultrasonic waves, cavitation bubbles oscillate, grow, and burst, resulting in instantaneous high temperature and pressure. The rapid alternation of temperature and pressure directly damages the cell wall or membrane of microorganisms, promotes water molecule decomposition, and triggers free radical reactions. These generated free radicals possess strong oxidizing properties that damage the DNA and enzymes of microbial cells, leading to microbial deactivation [23]. However, the penetration ability of ultrasonic waves is limited; therefore, it is commonly combined with other bacterial-reducing technologies in practical applications.

Irradiation sterilization technology radiates food utilizing ionizing radiation sources of various wavelengths, such as electron beams, γ-rays, or X-rays [24], reducing or removing most of the harmful microorganisms in food. This technology can minimize the decline in food quality, extend shelf life, and is widely used in the field of food processing and preservation. After irradiation, microorganisms in food (such as bacteria, yeast, and mold) absorb radiation energy, which breaks chemical bonds and changes cytochemical composition to achieve sterilization [24]. In general, to ensure the safety of processed meat, the maximum absorbed dose of irradiated food should not exceed 10 kGy according to the standards established by the Codex Alimentarius Commission (CAC) [25]. However, different countries have varying standards for using irradiation in food. In the food industry, it is also necessary to determine the irradiation dose based on different types of food in order to effectively kill harmful microorganisms while maintaining food quality. The high cost of setting up and maintaining irradiation facilities may limit their use in certain areas. Furthermore, the use of irradiation necessitates professional equipment and skilled personnel. Factors such as cost, technology, and regulations need to be comprehensively considered. With the advancement of technology and improvement in public awareness, irradiation technology is expected to be applied in more fields.

Ultraviolet, a non-ionizing radiation, has been approved by the FDA as a cold sterilization technology for reducing the microbial content of food. The wavelength of the UV emission spectrum ranges from 100 nm to 400 nm, which is longer than that of X-rays but shorter than that of visible light. According to the wavelength range, it can be divided into long-wave ultraviolet (UVA, 315~400 nm), medium-wave ultraviolet (UVB, 280~315 nm), and short-wave ultraviolet (UVC, 100~280 nm). Among them, UVC is the most commonly used ultraviolet range in food processing and has a high single-particle energy. It can kill bacteria, fungi, viruses, and other microorganisms by altering the permeability of microbial cell membranes and damaging microbial DNA or RNA [26]. In the process of sterilization, UV does not need to add any chemical substances, nor does it leave any harmful residues in food [27]. The penetration of ultraviolet light, however, is weak, and it can only sterilize the directly exposed surface. It belongs to ionizing radiation and requires attention to safety protection measures when used.

Cold plasma is a type of electroneutral ionized gas, which is composed of electrons, positive and negative ions, reactive oxygen species (ROS), excited or non-excited gas molecules, and photons. It is under the action of heat, an electric field, and microwaves so that some atoms or molecules are ionized to form substances based on reactive oxygen, reactive nitrogen, and ultraviolet photons. These substances interact with the food surface and cause microbial death. The production process of plasma involves high-energy electrons, and strict safety measures are required during operation to avoid injury to the operator. Compared with traditional heat treatment or chemical treatment technologies, the technical maturity and market application of cold plasma sterilization technology are still developing and need further verification and improvement.

High-pressure processing involves placing food in a fluid medium after soft packaging at room temperature or low temperature and subjecting it to pressures ranging from 100 to 1000 MPa for a certain period of time. This allows the pressure to be uniformly transmitted to the food [28], resulting in lethal effects on microorganisms. The purpose of sterilization is to affect gene expression, destroy the cytoplasmic membrane, denature proteins, and inactivate metabolic enzymes. Since covalent bonds are generally not broken by HPP, the aroma compounds of vitamins or foods are usually unaffected [29]. In contrast, large molecules, such as carbohydrates and proteins, undergo structural changes under stress. Additionally, it can extend the shelf life of food and maintain its nutritional value [30]. Therefore, HPP has a promising market application prospect.

### 2.3. Biological Bacteria Reduction Technology

Natural antimicrobial agents refer to a class of bactericidal substances with complex structures that are extracted from plants and animals in nature through bioengineering technologies or produced through microbial metabolism [31] and are widely used in meat preservation. According to its main sources, it can be divided into three categories: plant-derived natural antimicrobials, animal-derived natural antimicrobials, and microbial-derived natural antimicrobials.

Plant-derived antimicrobials are currently the most widely used natural antimicrobials in the market. They are substances with antibacterial activity extracted or synthesized from various tissues, roots, stems, leaves, fruits, seeds, and other parts of plants. They contain a variety of bioactive ingredients, such as volatile oils, alcohols, phenols, alkaloids, and enzymes. These ingredients can inhibit microorganisms through various mechanisms, thereby achieving a bacteriostatic effect.

Animal-derived bacteriostatic agents refer to substances with bacteriostatic activity extracted or synthesized from animals, which are generally obtained from various tissues, secretions, or bioactive substances synthesized through bioengineering technologies. Proteins and enzymes, such as chitosan, lysozyme, lactoferrin, propolis, and antimicrobial peptides, are the most common natural antimicrobial compounds found in animals. They have a wide range of killing effects on bacteria, fungi, and viruses while effectively inhibiting enzyme activity. As a result, they improve the safety and shelf life of food.

Natural antimicrobials derived from microbial sources are mainly obtained from various microorganisms, including bacteria, fungi, and some antimicrobial substances that are extracted or synthesized from microorganisms using biotechnology. The antibacterial action of specific agents can vary depending on the microbial targets. For example, some antibacterial agents produced by microbes themselves can interfere with microbial metabolism or produce antibiotics and enzymes. Additionally, competitive exclusion mechanisms can inhibit the growth and reproduction of other microorganisms.

### 2.4. Mechanisms of Bacterial Inhibition in Meat Reduction Technology

During the process of meat storage, meat is susceptible to the action of endogenous enzymes and microorganisms, which can lead to deterioration and degradation of quality. The main foodborne microorganisms that exist and contribute to meat spoilage include *Escherichia coli*, *Salmonella typhimurium*, *Bacillus cereus*, *Staphylococcus aureus*, *Listeria*, etc. At present, the commonly used meat bacteria reduction technology is mainly divided into chemical bacteria reduction technology, non-thermal physical bacteria reduction technology, and biological bacteria reduction technology. The characteristics (Table 1), bacteriostasis mechanisms (Table 2), and the diagram illustrating the bacteriostasis mechanism (Figure 1) of different bacteriostasis technologies are as follows.

## 3. Application of Bacteriological Reduction Technologies in Meat

### 3.1. Application of Chemical Bacteria Reduction Technology in Meat

#### 3.1.1. SAEW

The bactericidal effect of SAEW varies slightly depending on the treatment method used, such as soaking, atomization, spraying, and washing. Among these methods, soaking treatment yields the best bactericidal effect. Rahman et al. [41] soaked fresh chicken breast with SAEW, and this significantly reduced the number of natural flora and pathogenic bacteria and slowed down the rate of spoilage during refrigeration. Similarly, Sheng et al. [42] discovered that soaking beef in SAEW was effective in killing microorganisms and delaying the deterioration of refrigerated beef quality. Furthermore, the bactericidal effect of SAEW varies depending on the duration and concentration of treatment. Ye et al. [43] investigated the effects of different treatment times and concentrations of SAEW on raw frozen shrimp and found that treating with SAEW for 5 min had a better bactericidal effect than treating with SAEW for 3 min. Additionally, under the same treatment time, the bactericidal effect of 29 mg/L of SAEW was significantly enhanced compared to that of 16 mg/L of SAEW. However, SAEW has a limited ability to maintain oxidative stability during meat storage. Therefore, combining SAEW with antioxidants can help enhance its oxidative stability. However, further research is needed. In recent years, SAEW ice has been studied and applied for the preservation of meat and aquatic products [44]. The storage and preservation of aquatic products with ice-containing bactericidal substances can not only inhibit the reproduction of microorganisms on the surface of the products, but also extend their shelf life.

Currently, although SAEW technology has been widely utilized in the domestic and international food industries, there is still a lack of systematic research and evaluation on the sterilization principle of SAEW, its influencing factors, and its impact on the oxidation of different types of meat. This deficiency also hinders the promotion and application of SAEW in meat processing and industrialization. Maximizing the sterilization and preservation effects of SAEW, as well as exploring its combination with other technologies, remains the focus of future research.

#### 3.1.2. Organic Acids

The bacterial reduction effect of organic acids is closely related to their species, concentration, pH, volume, type of target microorganisms, and temperature of the carcass surface. Organic acids are often sprayed on meat to reduce bacteria. However, soaking the meat in organic acids can lead to cross-contamination and significant loss of soluble substances, which affects the effectiveness of bacteria reduction. Some scholars have explored the bacteria-reducing effect of nine different organic acids (including lactic acid, acetic acid, citric acid, and peracetic acid) on cattle carcasses and found that lactic acid is the most effective measure for reducing bacteria. It can effectively reduce the total number of colonies and coliform bacteria [45]. In addition, lactic acid is the most commonly used organic acid in the meat industry to reduce bacteria in products due to its effectiveness and cost. Furthermore, the European Union permits the spraying of 2% to 5% lactic acid on carcasses as a means of reducing bacteria [46]. Ransom et al. [47] demonstrated that applying 2% lactic acid can decrease *E. coli* O157:H7 on cattle carcass surfaces by 1.6 log CFU/g. Manzoor et al. [48] also compared the effects of different concentrations of lactic acid spray on the microbial and sensory characteristics of buffalo meat and found that both could significantly reduce the number of microorganisms, while a concentration of lactic acid above 6% would adversely affect the color of the meat. When a higher temperature (above 55 °C) is combined with a lower concentration (2%) of lactic acid spray, the bactericidal effect is enhanced. However, the stability of organic acids is compromised under high-temperature conditions, leading to changes in meat’s sensory properties and the emergence of acid-resistant pathogens. Additionally, this may also result in the corrosion of processing equipment [45]. The study found that a combination of two or more organic acids is more effective than a single organic acid, which can enhance the bactericidal effect and overall food quality. Surve et al. [49] found that combining acetic acid with lactic acid or propionic acid can significantly enhance the antimicrobial properties of buffalo meat without affecting its color and flavor, effectively extending its refrigerated shelf life.

In the future, new organic acids or derivatives of organic acids could be developed to address the issues of heat resistance and broad-spectrum antimicrobial properties associated with existing organic acids. Despite some limitations in food sterilization, organic acids have broad application prospects as a natural and environmentally friendly sterilization technology due to increasing consumer demand for natural and safe foods.

#### 3.1.3. Ozone

The application of ozone to meat can be divided into two forms: gaseous state and aqueous solution [12]. Its safety and effectiveness have been proven to kill a variety of foodborne pathogens in food, such as *Escherichia coli*, *Salmonella*, and *Listeria monocytogenes* [50]. Cárdenas et al. [51] analyzed the antibacterial effect of gaseous ozone on chilled fresh beef and found that an ozone concentration of 141.12 mg/m^3^ could reduce the total number of *E. coli* and inoculated microorganisms, resulting in complete inactivation of certain microorganisms and maintaining meat freshness. The bactericidal effect of ozone is influenced by factors such as its concentration, temperature, contact time, and food characteristics. Stivarius et al. [52] investigated the bactericidal effect of a 1% ozone aqueous solution and found that a 7 min ozone treatment had a better bactericidal effect than a 5 min treatment. Additionally, researchers compared a high ozone dose (1000 ppm) with a low ozone dose (100 ppm) and discovered that although the high ozone dose can slow down microbial activity on the surface of pork and reduce physiological activity, it is not sufficient to have a fatal effect on microbial life in meat. This may be attributed to the longer incubation times (46 and 49 h) of the samples under non-sterile conditions after ozone treatment [53]. However, ozone is also a potent oxidizing agent that can impact the sensory, physical, and chemical properties of meat and meat products. Ayranci et al. [54] discovered that ozone treatment (1 × 10^−2^ kg m^−3^, 8 h) had detrimental effects on the physicochemical properties and color of turkey breast. The high oxidation potential of ozone may be the primary reason for lipid and protein oxidation in meat. Additionally, it was observed that the connective tissue membrane with high protein content in turkey meat underwent denaturation after ozone treatment, which could explain the lighter coloration of turkey meat.

Ozone sterilization systems require complex equipment and precise control, making them costly to operate and maintain. Furthermore, high concentrations of ozone can be toxic to the human body, so it is necessary to strictly regulate the concentration and contact time during actual operation in order to ensure food safety and protect operators.

### 3.2. Application of Non-Thermal Physical Bacteria Reduction Technology in Meat

#### 3.2.1. Ultrasound

In the food industry, high-power ultrasonic waves with a frequency of 20~100 kHz are generally used to inactivate microorganisms [55]. The sterilization effect of ultrasonication is mainly influenced by the ultrasonic medium, frequency, action time, and microbial species [56]. Huu et al. [57] used ultrasonic treatments with a frequency of 40 kHz and a power density of 0.092 W/mL for *E. coli* O157:H7 or *L. innocua*. The initial bacterial content was 1 × 10^6^ CFU/mL, and the bacterial content remained at the initial level after both 30 min and 45 min of ultrasonic treatment. The results showed that there was no significant influence on either bacteria in the product during the entire period of ultrasonic treatment. It has been found that a combination of ultrasonic and heat treatments can accelerate the sterilization speed of food. Morild et al. [58] evaluated the effect of pressurized steam combined with high-power ultrasound on pathogen inactivation on pig skin and pork surface. The study examined the inactivation of *Salmonella typhimurium*, *Salmonella* Derby, *Salmonella* Infantile, *Yersinia enterocolitica*, and a non-pathogenic *E. coli*. After 4 s of ultrasound treatment, the total number of colonies decreased by 3.3 log CFU/cm^2^. Musavian et al. [59] also found that steam treatment and ultrasonic treatment of chicken carcasses on the processing line could significantly reduce the number of *Campylobacter* on contaminated poultry. By using steam and ultrasound immediately after slaughter, the total count of viable bacteria was reduced by approximately 3 log CFU/cm^2^. However, prolonged exposure of food to high temperatures at ultrasonic wavelengths leads to a reduction in its functional properties, sensory properties, and nutritional value [60]. Therefore, it can be combined with other technologies that reduce bacteria to enhance its bactericidal effect. Kordowska-Wiater et al. [61] investigated the isolation of *Salmonella enterica* ssp. *enterica* sv. Anatum, *Escherichia coli*, *Proteus* sp., and *Pseudomonas fluorescens* from the surface of chicken skin after being treated with ultrasound (at a frequency of 40 kHz, intensity of 2.5 W/cm^2^, and duration of either 3 or 6 min) in water and a 1% lactic acid solution. It was observed that treating chicken skin with ultrasound alone in a lactic acid solution for 3 min resulted in a reduction of 1.0 log CFU/cm^2^ in the number of these microorganisms. Furthermore, extending the treatment time to 6 min led to a decrease exceeding 1.0 log CFU/cm^2^ in the number of microorganisms present in the water sample.

Ultrasound has varying effects on the inactivation of different microorganisms (bacteria, viruses, fungi, and mycotoxins) as well as food substrates. The frequency, intensity, and treatment time of ultrasonic waves need to be optimized for different types of food. Several studies have shown that while ultrasound can reduce the number of bacteria produced by meat, using two or more bacteria-reducing technologies is more effective than relying solely on bacterial reduction [21]. Therefore, it is rarely used alone in actual meat production processes for reducing bacteria and usually works in conjunction with other bacteria-reducing technologies.

#### 3.2.2. Irradiation

Irradiation technology has been approved by the FDA for use in food processing. Food irradiation is a form of “cold treatment” that effectively kills bacteria without significantly increasing the internal temperature or causing nutrient loss in the food [47]. Electron beam radiation (EB) and X-ray radiation (XR) are more acceptable to consumers than gamma rays (GR), because they do not involve radioactive isotopes [48]. However, Park et al. [62] found that GR irradiation may be more effective in reducing bacterial populations than EB irradiation. Additionally, doses of 5~10 kGy for GR and EB irradiation, respectively had no adverse effects on the lipid oxidation and sensory characteristics (color, chewability, and taste) of beef sausage patties. The technical advantages of high-energy X-rays include better power utilization, dose uniformity, and shorter irradiation times. Consequently, the use of X-ray irradiation results in higher productivity and lower processing costs. Yim et al. [63] used X-ray irradiation to treat beef samples, and as the irradiation dose increased, there was a significant decrease in the total number of aerobic bacteria present. No bacterial growth was observed in the samples treated with a dose of 10 kGy. The combined treatment of irradiation and natural antimicrobial agents in raw meat materials can enhance the degree of inhibition of foodborne pathogens and extend the shelf life of meat and meat products. Hu et al. [64] combined chitosan-eugenol with irradiation to significantly reduce the number of *Staphylococcus aureus* and *Salmonella* in fresh pork, as well as delay fat oxidation during processing.

However, although meat and its products treated with low-dose radiation will not be contaminated by radioactivity and play a good role in inhibiting the growth of pathogenic microorganisms in meat, it is important to select the appropriate radiation for each type of food during use. Additionally, parameters such as radiation dose and time should be well controlled to minimize adverse effects on the quality of meat flavor, color, nutrition, and moisture [65].

#### 3.2.3. Ultraviolet

UV sterilization technology has the advantages of a broad-spectrum, high efficiency, and no secondary pollution. It has been proven to effectively reduce pathogenic microorganisms on the surface of meat products and extend their shelf life. It is widely used for meat preservation. In the study conducted by Söbeli et al. [66], various doses of UVC were used to irradiate beef fillet, and it was observed that a higher dose (4.2 J/cm^2^) significantly reduced the total number of aerobic bacteria by 3.49 ± 0.67 log CFU/g. In contrast, Bryant et al. [67] investigated the effect of UV radiation on the inactivation of *E. coli* K12 on beef surfaces. It was shown that microbial inactivation by UV treatment was correlated with both the duration of treatment and the distance between the sample and the light source. Generally, the longer the treatment time and the shorter the distance between UV and the sample, the higher will be the rate of microbial inactivation. The bactericidal effect of ultraviolet light varies depending on the specific type of microorganism. McLeod et al. [68] investigated the inactivation of pathogenic bacteria in chickens treated with continuous UVC. The results demonstrated that exposure to 3.0 J/cm^2^ of UVC reduced the log CFU/cm^2^ counts of *Carnobacterium divergens,* enterohemorrhagic *E. coli* and *Pseudomonas* spp. by 2.8, 1.7, and 2.7 log CFU/cm^2^ respectively. However, due to the poor penetration ability of UV in reaching the interior of the food, it can only reduce the survival rate of the initial bacterial count on the surface of the meat sample and cannot eliminate all microorganisms. Therefore, it is often used in combination with other sterilization technologies to enhance its effectiveness. Studies have found that the use of UV in combination with ozone [69] or other methods, such as peracetic acid and lactic acid [70], can effectively eliminate microorganisms in meat. At the same time, meat may undergo a series of oxidation processes after being exposed to a high dose of UV treatment, which can potentially affect the quality of the product. To mitigate the UV oxidation of meat lipids and proteins, hurdle technology can be employed. For example, UV can be combined with deaerator packaging, vacuum packaging, and high hydrostatic pressure [71,72]. Therefore, the development of diverse UV combination technologies is essential to ensure the quality and safety of meat for future UV applications.

Compared to fruits and vegetables, UV is used in meat and meat products to a relatively small extent. The bactericidal effect of different UV irradiation doses on meat varies. However, there is currently limited research on the relationship between dose and meat disinfection. Additionally, the application of ultraviolet in meat needs further exploration regarding appropriate wavelength, dose, and luminescence methods for different types of meat. Therefore, future work should focus on expanding basic research on UV in meat and meat products, as well as researching and developing new UV technologies.

#### 3.2.4. Cold Plasma

In recent years, cold plasma has gained widespread attention in the field of food sterilization and has shown great potential for meat preservation applications. It can effectively eliminate various types of viruses, bacteria, fungi, and spores. The ability of cold plasma to inhibit bacteria mainly depends on factors such as gas composition, airflow, electrical input, and duration of the process [73]. Kim et al. [74] used helium (10 lpm) and a mixture of helium and oxygen (10 lpm and 10 sccm) as the excitation medium to process sliced bacon under the conditions of input power at 75, 100, and 125 W, with treatment times of 60 and 90 s respectively. After plasma helium treatment and helium/oxygen mixed treatment, the total number of aerobic bacteria decreased by 1.89 and 4.58 log CFU/g, respectively. The study also found that regardless of the gas composition, the bacteria-reducing effect was enhanced with increasing input power and treatment time. In addition, low-temperature plasma has different effects on reducing bacteria in different microbial species. Choi et al. [75] used a 20 kV direct current with an output voltage of 58 kHz frequency. It was found that the optimal condition for microorganism inactivation was achieved when the plasma was generated at a current of 1.5 A and the distance between the plasma electrode tip and the sample was 25 mm. After treating pork samples for 120 s, it was observed that *E. coli* O157:H7 had a stronger inactivation effect compared to *Lactobacillus monocytogenes*, and there were no significant changes in pork color, flavor, nutrition, or other quality indexes after treatment. Moreover, Ulbin-Figlewicz et al. [76] conducted a study on the treatment of helium and argon plasma to compare their effects on inactivating microflora on the surfaces of different types of meat. Under an operating pressure of 20 kPa, after 10 min of helium plasma treatment, the total number of colonies in pork decreased by 1.14 to 1.48 log CFU/cm^2^, while that in beef only decreased by 0.98 to 2.09 log CFU/cm^2^.

Currently, technology plays a significant role in the sterilization and preservation of meat and meat products; however, it is still in the early stages of basic research with some issues such as weak penetration ability. When a large number of microorganisms gather on the food surface, resulting in uneven bactericidal effects, its effectiveness will be affected. The technology can be combined with other non-heat treatment methods to enhance its bactericidal effect. Additionally, there are challenges like high equipment investment costs. The specific influencing factors, mechanism of action, and regulatory means of this technology lack a well-established theoretical foundation for support and require further exploration in future studies to promote its wider application in the meat industry.

#### 3.2.5. HPP

The effectiveness of HPP for microbial control depends on various factors, including food composition, types of microbes present, the method and magnitude of applied pressure, duration of pressure application, treatment temperature, and the integrity of packaging [77]. Hayman et al. [78] conducted a study using HPP on different types of meat (low-fat pastrami, Strasbourg beef, export sausage, and Cajun beef), which demonstrated that treatment at 600 MPa and 20 °C for 180 s extended the refrigerated shelf life of ready-made meats. The number of *L. monocytogenes* in the inoculated products was reduced by more than 4 log CFU/g. In order to enhance the bactericidal effect of HPP, antibacterial agents can be combined with HPP. Melhem et al. [79] investigated the combined inactivation effect of HHP on pathogenic and spoilage bacteria in meat products using different application methods (surface application, product incorporation, and active packaging) and various types of antibacterials (derived from plants, microorganisms, and animals). It was discovered that combining bacteriocin with HHP resulted in a stronger bacteriocidal effect due to its significant inhibitory impact on Gram-positive bacteria like Listeria. Regarding application methods, the synergistic effect of active packaging and HHP proved to be the most effective approach for pathogenic bacteria inactivation in meat products. However, the use of HPP may have an effect on the color of fresh meat, and the exact mechanism of color change is unknown. This could be attributed to the disruption of non-covalent bonds, protein denaturation, and myoglobin oxidation in fresh meat under high pressure [77,80]. Studies have shown that compared with beef or mutton treated with HPP, poultry meat has a lower myoglobin content and is less affected by HPP treatment [81]. To investigate this issue, Gupta et al. [80] discovered that altering the myoglobin status can reduce the discoloration caused by HPP. Patties were prepared using different packaging methods (high oxygen–oxymyoglobin and carbon monoxide–carboxymyoglobin) or through the addition of potassium ferricyanide (metmyoglobin). It was found that carboxymyoglobin exhibited better color retention.

The application of HPP as a non-thermal physical technology for meat bacterialization has shown great potential; however, it still faces some challenges. For instance, bacterial spores are highly resistant to high pressure [82] and require higher pressure (>1200 MPa) [28] for their inactivation. However, these higher-pressure conditions may have adverse effects on food quality. Therefore, in the future, it can be used in combination with other methods to reduce bacteria and sterilize food.

### 3.3. Application of Biological Bacteria Reduction Technology in Meat

Natural antimicrobials have long been used as food additives, primarily to extend the shelf life and maintain the quality of meat and meat products. Among natural antibacterial compounds, plant-derived antimicrobials have been extensively studied, and substances such as phenols, isoflavones, ketones, acids, terpenes (essential oils), and alkaloids commonly found in meat can limit or hinder the growth of harmful microorganisms. Different types of microorganisms in meat and meat products are inhibited by various plant extracts. Studies have shown that when thyme and balm essential oils [83] are applied to chicken breast, balm essential oil significantly restricts the growth of *Salmonella* sp., while thyme essential oil effectively inhibits the growth of *E. coli.* Furthermore, they are incorporated into meat in various forms, such as direct mixing, coating, layering, nanoencapsulation, and microencapsulation. This can enhance its antibacterial properties and significantly expand the application of natural antimicrobials in meat. For instance, Noshad et al. [84] blended *Citrus limon* essential oil with psyllium seed slime to create a novel edible coating for beef storage and preservation and discovered that it could extend the shelf life by 10 days. Moreover, different addition technologies may have varying inhibitory effects on microorganisms. This was demonstrated by Wang et al. [85], who found that nanoemulsions of *Litsea cubeba* essential oil exhibited stronger inhibition against *L. monocytogenes* and spoilage bacterial *S. baltica* compared to pure *Litsea cubeba* essential oil.

Chitosan, a polycationic biopolymer commonly extracted from the exoskeletons of crustaceans, such as crabs and lobsters, is frequently used as an animal-derived antimicrobial agent in meat. Darmadji et al. [86] discovered that chitosan at concentrations ranging from 0.1% to 1% effectively inhibited the growth of spoilage bacteria and pathogenic bacteria during the storage of refrigerated meat. In addition to chitosan, lysozyme extracted from eggs and milk is also commonly employed. Furthermore, lactoferrin [87], a natural antibacterial substance found in secretions like saliva, milk, and tears of mammals, can inhibit the activity of *E. coli.* It is also possible to produce lactoferrin nanoparticles by electrostatically chelating gellan gum [88], thereby enhancing their antimicrobial capacity without requiring inorganic compounds. This greatly expands the utilization of lactoferrin as a natural antimicrobial agent in food.

There are various types of microorganisms, and microbial-derived agents that reduce bacteria have gradually become the focus of research. Bacteriocins are typically produced by both Gram-positive and Gram-negative bacteria [2]. Studies have shown that lactic acid bacteria isolated from buffalo milk curds produce bacteriocins with antibacterial properties that can inhibit the growth of pathogenic microorganisms in chickens [89]. Another bacteriocin commonly used in meat is nisin, which is produced by certain strains of *Lactococcus lactis* and has the ability to inhibit various Gram-positive bacteria [90]. Arief et al. [91] achieved a significant reduction in the growth of *Escherichia coli* in mutton intestines by adding probiotic *Lactobacillus plantarum* IIA-2C12, which also preserved the color of meat products. However, this addition also resulted in changes in the taste of the meat and decreased acceptability to some extent. Therefore, investigating the impact of microbial preservatives on the flavor of meat products is an important area for future research. Araújo et al. [92] also investigated the bacterioreducing effects of a combination of garlic essential oil (GO), allyl isothiocyanate (AITC), and nisin (NI) on fresh intestines. The combination of 20 mg/kg NI + 125 μL/kg GO + 62.5 μL/kg AITC or 20 mg/kg NI + 62.5 μL/kg GO + 125 μL/kg AITC was found to have a significant inhibitory effect on *E. coli* O157H7 and spoilage lactic acid bacteria, which was better than that of single applications, without affecting the sensory acceptability of fresh intestine.

However, in practical applications, the development and utilization of natural antimicrobial agents are still imperfect, with problems such as instability, insufficient extraction of available active substances, and limited effectiveness of a single antimicrobial agent, which may prevent achieving the expected results. To better utilize natural antimicrobials, new composite antimicrobials can be synthesized by optimizing their extraction and purification methods and utilizing new technologies [85] to enhance their stability and antibacterial effects. This will facilitate their improved application in the field of meat preservation.

### 3.4. Application of Hurdle Technology in Meat

The above bacteriological reduction technologies can be used to improve the safety and shelf life of food ingredients. However, the antibacterial effects of some of these technologies are not obvious when used alone, and they adversely affect the organoleptic properties of foods and reduce consumer acceptability. Owing to these facts, the hurdle concept (generally known as combined methods, combination preservation, combined processes, barrier technology, or combination techniques) has become a promising technology that simultaneously reduces losses of nutritional and sensory quality and improves food safety [41]. The hurdle technology is a method for ensuring the safety of food products by controlling or destroying microbial spoilage or food pathogens through the combination of a series of preservative parameters (hurdles) imposing physiological impact on microbial cells. Early studies have shown that the preservation period of foods with low initial bacterial counts can be extended 1~2 times longer than those with high initial bacterial counts. If the initial amount of bacteria inside the food is low, only a few defense factors are needed for effective bacterial inhibition; conversely, if poor sanitary conditions cause a high initial amount of bacteria, it is necessary to increase the defense factor or enhance its strength.

Many researchers utilize physical sterilization technology, chemical bactericidal technology, biological bactericidal technology, and low-temperature preservation technology as a combination of defense factors to achieve the preservation effect of “1 + 1 > 2” through hurdle technology. Mikš-Krajnik et al. [6] pointed out that SAEW alone was not sufficient for complete inactivation of microorganisms. However, combining SAEW with ultrasound [7] has been shown to provide rapid and effective bactericidal results. Similarly, ozone has been combined with defense factors such as irradiation [9], organic acids [8], and ultraviolet light [93] to control microbial growth and extend the shelf life of food. During the process of meat storage, appropriate techniques for reducing bacteria are scientifically determined and selected based on various factors such as ventilation, as well as considering factors like the type of meat, processing methods, and storage conditions. In addition, by following the correct dosage and handling methods to ensure its effectiveness and safety, achieving the quality of meat storage becomes easy. With the rapid development of international food science, technology, and industry, the research and application of hurdle technology will become increasingly extensive and in-depth, providing a reliable theoretical basis for future meat preservation.

## 4. Conclusions

This review summarizes the characteristics, principles, and applications of commonly used bactericidal technologies in meat. Each individual bactericidal technology has its own characteristics and can achieve the expected effect to a certain extent. However, there are still limitations, as the bactericidal effect is easily restricted by different objective factors. Chemical bactericidal technology is widely used due to its high efficiency; however, frequent use of the same chemical agent may lead to microbial resistance. Non-thermal physical bacteria reduction technology causes less damage to food color, flavor, and nutrition. It does not cause pollution. However, it typically requires a specific environment and equipment, resulting in high energy consumption. Additionally, the killing effect on microorganisms may be unstable, especially when there are significant changes in environmental conditions or complex microbial species present. As a result, this can affect the overall effectiveness and comprehensiveness of the technology. Biological bacteria reduction technology is safe, utilizes a wide range of biological resources, and exhibits high levels of biodiversity and adaptability. However, it is easily influenced by the extraction rate, climate, geographical conditions, and processing costs. With continuous development and in-depth research on bactericidal technology based on the “hurdle technology” principle, different bactericidal pretreatment measures are combined according to the type of food. This targets microorganisms in meat raw materials for control purposes, thereby enhancing meat quality and ensuring food safety. These advancements will shape the future trend of meat preservation while facilitating its industrialization and application.

## Figures and Tables

**Figure 1 foods-13-02361-f001:**
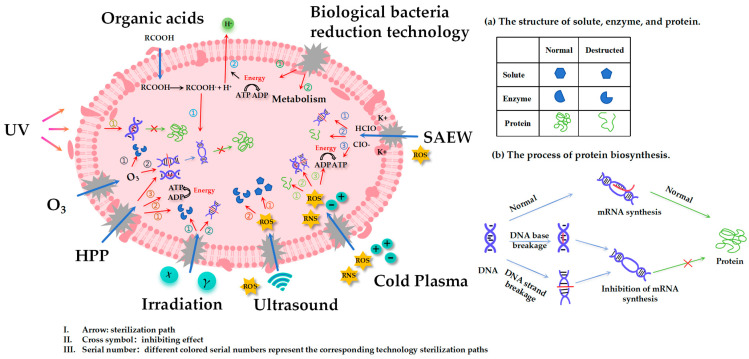
Mechanism of bacteria-reducing technology.

**Table 1 foods-13-02361-t001:** The classification and characteristics of bacteria-reducing technology.

Classification		Advantage	Disadvantage
Chemical bacteria reduction technology	SAEW	Efficient sterilization, convenient manufacturing, low cost, wide application, safety and environmental protection	Effect instability
	Organic acids	Low cost, green, efficient sterilization	Unstable and easy to decompose
	Ozone	Efficient sterilization, no secondary pollution	High equipment cost and poor stability
Non-thermal physical bacteria reduction technology	Ultrasound	Green safety, wide applicability, convenient and fast	Limited penetration, uneven sterilization
	Radiation	Efficient sterilization, cold treatment, no residue, strong controllability, wide application, environmental protection, and energy-saving	High cost, high equipment, and technical requirements
	UV	Environmentally friendly, no residue	High energy consumption, safety risks, penetration limitations, and environmental impacts
	Cold plasma	Mild, efficient, no residue	Complex technology, poor stability, technical maturity
	HPP	High sterilization efficiency, low energy consumption, green and safe	High equipment cost and application limitation
Biological bacteria reduction technology	Plant-derived natural antimicrobials	Environmentally friendly, green and safe	Obviously seasonal and regional, the effect is affected by the separation and extraction process
	Animal-derived natural antimicrobials	Natural origin, broad-spectrum, safety, biocompatibility, and ease of application	Drug resistance and antimicrobial function are limited
	Microbial-derived natural antimicrobials	Wide range of sources, high security	Effect instability

**Table 2 foods-13-02361-t002:** Mechanism of bacteria-reducing technology.

Bacteria Reduction Technology	Categorization	Mode of action	Antibacterial Mechanism
Chemical bacteria reduction technology	SAEW	HCIO CIO^−^ ROS	The cell membrane is damaged by SAEW, causing rapid leakage of K^+^ and an increase in membrane permeability. This causes HCIO and CIO^−^ to enter the cell, resulting in the following consequences: (1) It causes the breakage of DNA base, which inhibits the protein biosynthesis; (2) It causes the breakage of protein structure; (3) It causes changes in electron flow, inhibiting microbial energy metabolism and ATP production [32].
	Organic acids	RCOOH COOH^−^	RCOOH enters into the cell, leading to the following consequences: (1) RCOOH ionized by RCOOH causes the breakage of DNA base and DNA strand, which inhibits protein biosynthesis; (2) The H^+^ ionized by RCOOH reduces the pH value of the cells, and H^+^ is excreted out of the cell through energy released by the transformation from ATP into ADP. The consumption of a large amount of energy inhibits the reproduction of bacteria.
	Ozone	O_3_	O_3_ increases the permeability of the cell membrane and destroys lipoproteins and lipopolysaccharides, resulting in the following results after entering the cell: (1) It causes the breakage of enzyme structure; (2) It disrupts the structures of DNA and RNA, interfering with protein synthesis [19,33].
Non-thermal physical bacteria reduction technology	Ultrasound	Cavitation effect ROS	Microbial cells experience violent oscillations that disrupt the permeability of cell membranes and release reactive oxygen species enter the cell, which results in the following consequences: (1) The water molecules in the liquid medium decompose into reactive oxygen species, causing reduction reactions and secondary oxidation, which results in changes to the structure of the solute [34]; (2) It causes the breakage of enzyme structure [35].
	Radiation	Electron rays γ-rays X-rays	(1) Electron rays damage the inner cell membrane, leading to disruption of the enzyme system and impairment of functions; (2) It causes the breakage of the DNA bases of microorganisms [36].
	UV	UV	It destroys the DNA base, inhibits DNA transcription, replication, and cell division, and inhibits protein synthesis by altering or destroying the structure of DNA or RNA molecules [1].
	Cold plasma	Active substances ROS Electrically charged particles	ROS, reactive nitrogen species (RNS), and charged particles destroy bacterial cells and then enter the cell interior, resulting in the following consequences: (1) It causes the destruction of protein structure; (2) It causes the breakage of DNA bases and DNA strands of microorganisms; (3) It promotes the transform from ADP into ATP, resulting in energy consumption.
	HPP	High pressure	HPP destroys cell membranes and leads to cytoplasm loss [37], resulting in the following consequences: (1) It induces enzyme denaturation, resulting in the destruction of enzyme structure [38]; (2) It causes the destruction of ribosomes and promotes the transform from ADP into ATP, resulting in energy consumption [39]; (3) The nucleic acids (DNA and RNA) inside the cell are damaged and disruption of protein biosynthesis [40].
Biological bacteria reduction technology	Plant-derived natural antimicrobials	Plant antimicrobials Volatile components Tannates Aromatic compounds	It enters the cell through diffusion to disrupt microbial cell walls and cell membranes, inhibiting ATP synthesis and reducing energy metabolism
	Animal-derived natural antimicrobials	Amino acids Polymer sugars	Damage to cell walls and cell membranes results in increased membrane permeability, which affects energy conversion and synthesis of biomolecules and disrupts cell metabolism.
	Microbial-derived natural antimicrobials	Microbial metabolites	It alters the permeability of cell membranes or inhibits the growth of microorganisms through competition for nutrients.

## Data Availability

No new data were created or analyzed in this study. Data sharing is not applicable to this article.

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
