# Peer review of "Research Progress on Bacteria-Reducing Pretreatment Technology of Meat"

_foods, 2024, doi:10.3390/foods13152361_

Round 1

Reviewer 1 Report

Comments and Suggestions for Authors

a)The review is too general and does not contain significant scientific information, neither summary from the literature.

b) did authors used ai for manuscript writting?

c) Authors need to include 2 paragraphs minimum for each sterilization technique, summarizing published reports.

d) Also comparison between sterilization yechniques, in terms of advantages/disadvantages is missing.

e) Content of review is missing.

Author Response

Dear Editors and Reviewers:

Thank you for your letter and for the reviewers’ comments concerning our manuscript entitled “Research Progress on Bacteria-Reducing Pretreatment Technology of Meat”. Thanks to the experts for reviewing our article and giving us honest advice to improve the manuscript. Thank you very much for giving us the opportunity to rework our work and improve the manuscript. You are right that much revision and clarification need to be done in order to paper might possibly be published. Thanks again.

We consider the comments of all three reviewers very carefully and revise the manuscript according to the reviewers comments one by one.

The manuscript has been revised carefully according to your and the reviewers' comments. In our rebuttal indicate the line number in the revised manuscript corresponding to each change that has been made and use yellow highlighting in the text to indicate the edits. We have carefully made correction which we hope to meet with approval. Changes are marked in red in the manuscript. The responds to the reviewers’ comments and all corrections in the paper are as follow:

Response to Reviewer #1

Thank you very much for the recognition of this study. We would like to express our most sincere appreciation for your careful review of this manuscript. The valuable suggestions you had proposed were beneficial to improve the quality of this manuscript. The modifications have been performed carefully according to your suggestions as follow:

Q1: The review is too general and does not contain significant scientific information, neither summary from the literature.

A1: Thanks for your review carefully. In the revised manuscript, the comparison of the advantages and disadvantages among these technologies, the parameters of the technologies, and the summary of the technologies have been supplemented as follow:

  • In the revised manuscript, the sentences about the parameters of the technologies have been supplemented in line 265-559 as follow:

3.1.1. SAEW

Ye et al. [43] investigated the effects of different treatment times and concentrations of SAEW on raw frozen shrimp and found that treating with SAEW for 5 minutes had a better bactericidal effect than treating with SAEW for 3 minutes. Additionally, under the same treatment time, the bactericidal effect of 29 mg/L of SAEW was significantly enhanced compared to that of 16 mg/L of SAEW. However, SAEW has limited ability to maintain oxidative stability during meat storage. Therefore, combining SAEW with antioxidants can help enhance its oxidative stability. However, further research is needed. In recent years, SAEW ice has been studied and applied for the preservation of meat and aquatic products [44]. The storage and preservation of aquatic products with ice containing bactericidal substances can not only inhibit the reproduction of microorganisms on the surface of the products but also extend their shelf life.

3.1.2. Organic acids

In addition, lactic acid is the most commonly used organic acid in the meat industry to reduce bacteria in products due to its effectiveness and cost. Furthermore, the European Union permits the spraying of 2 % to 5 % lactic acid on carcasses as a means of reducing bacteria [46]. Ransom et al. [47]demonstrated that applying 2 % lactic acid can decrease E. coli O157:H7 on cattle carcass surfaces by 1.6 Log (CFU/g). Manzoor et al. [48] also compared the effects of different concentrations of lactic acid spray on the microbial and sensory characteristics of buffalo meat and found that both could significantly reduce the number of microorganisms, while a concentration of lactic acid above 6 % would adversely affect the color of the meat. When a higher temperature (above 55 ℃) is combined with a lower concentration (2 %) of lactic acid spray, the bactericidal effect is enhanced. However, the stability of organic acids is compromised under high temperature conditions, leading to changes in meat's sensory properties and the emergence of acid-resistant pathogens. Additionally, this may also result in corrosion to processing equipment [45]. The study found that the combination of two or more organic acids is more effective than a single organic acid, which can enhance the bactericidal effect and overall food quality.

3.1.3. Ozone

The bactericidal effect of ozone is influenced by factors such as concentration, temperature, contact time, and food characteristics. Stivarius et al. [52] investigated the bactericidal effect of a 1 % ozone aqueous solution and found that a 7-minute ozone treatment had a better bactericidal effect than a 5-minute treatment. Additionally, the researchers compared high ozone dose (1000 ppm) with low ozone dose (100 ppm) and discovered that although the high ozone dose can slow down microbial activity on the surface of pork and reduce physiological activity, it is not sufficient to have a fatal effect on microbial life in meat. This may be attributed to the longer incubation time (46 and 49 hours) of samples under non-sterile conditions after ozone treatment [53]. However, ozone is also a potent oxidizing agent that can impact the sensory, physical, and chemical properties of meat and meat products. Ayranci et al. [54] discovered that ozone treatment (1 x 10-2 kg m-3, 8 hours) had detrimental effects on the physicochemical properties and color of turkey breast. The high oxidation potential of ozone may be the primary reason for lipid and protein oxidation in meat. Additionally, it was observed that the connective tissue membrane with a high protein content in turkey meat underwent denaturation after ozone treatment, which could explain the lighter coloration of turkey meat.

Ozone sterilization systems require complex equipment and precise control, making them costly to operate and maintain. Furthermore, high concentrations of ozone can be toxic to the human body, so it is necessary to strictly regulate the concentration and contact time during actual operation in order to ensure food safety and protect operators.

3.2.1. Ultrasound

In the food industry, high-power ultrasonic waves with a frequency of 20-100 kHz are generally used to inactivate microorganisms [55]. The sterilization effect of ultrasonication is mainly influenced by the ultrasonic medium, frequency, action time, and microbial species [56]. Huu, et al.[57] used ultrasonic treatments with a frequency of 40 kHz and a power density of 0.092 W/mL for E. coli O157:H7 or L. innocua. The initial bacterial content was 1 × 106 CFU/mL, and the bacterial content remained at the initial level after both 30 min and 45 min of ultrasonic treatment. The results showed that there was no significant influence on either bacteria in the product during the entire period of ultrasonic treatment. It has been found that the combination of ultrasonic and heat treatment can accelerate the sterilization speed of food. Morild et al. [58] evaluated the effect of pressurized steam combined with high-power ultrasound on pathogen inactivation on pig skin and pork surface. The study examined the inactivation of Salmonella typhimurium, Salmonella Derby, Salmonella infantile, Yersinia enterocolitica, and a non-pathogenic E. coli. After 4 seconds of ultrasound treatment, the total number of colonies decreased by 3.3 log CFU/cm2. Musavian et al. [59] also found that steam treatment and ultrasonic treatment of chicken carcasses on the processing line could significantly reduce the number of campylobacter on contaminated poultry. By using steam and ultrasound immediately after slaughter, the total count of viable bacteria was reduced by approximately 3 log CFU/cm2. However, prolonged exposure of food to high temperatures at ultrasonic wavelengths leads to a reduction in its functional properties, sensory properties, and nutritional value [60]. Therefore, it can be combined with other technologies that reduce bacteria to enhance its bactericidal effect. Kordowska-Wiater et al. [61] investigated the isolation of Salmonella, Escherichia coli, Proteus, and Pseudomonas fluorescence from the surface of chicken skin after being treated with ultrasound (at a frequency of 40 kHz, intensity of 2.5 W cm2, and duration of either 3 or 6 minutes) in water and a 1% lactic acid solution. It was observed that treating the chicken skin with ultrasound alone in a lactic acid solution for 3 minutes resulted in a reduction of 1.0 log CFU/cm2 in the number of these microorganisms. Furthermore, extending the treatment time to 6 minutes led to a decrease exceeding 1.0 log CFU/cm2 in the number of microorganisms present in the water sample.

Ultrasound have varying effects on the inactivation of different microorganisms (bacteria, viruses, fungi, and mycotoxins) as well as food substrates. The frequency, intensity, and treatment time of ultrasonic waves need to be optimized for different types of food. Several studies have shown that while ultrasound can reduce the number of bacteria produced by meat, using two or more bacteria-reducing technologies is more effective than relying solely on bacterial reduction [21]. Therefore, it is rarely used alone in actual meat production processes for reducing bacteria and usually works in conjunction with other bacteria-reducing technologies.

3.2.2. Irradiation

Irradiation technology has been approved by the Food and Drug Administration (FDA) for use in food processing. Food irradiation is a form of "cold treatment" that effectively kills bacteria without significantly increasing the internal temperature or causing nutrient loss in the food [47]. Electron beam radiation (EB) and X-ray radiation (XR) are more acceptable to consumers than gamma rays (GR) because they do not involve radioactive isotopes [48]. However, Park et al. [62] found that GR irradiation may be more effective in reducing bacterial populations than EB irradiation. Additionally, doses of 5-10 kGy for GR and EB irradiation respectively had no adverse effects on the lipid oxidation and sensory characteristics (color, chewability, and taste) of beef sausage patties. The technical advantages of high-energy X-rays include better power utilization, dose uniformity, and shorter irradiation times.

3.2.3. Ultraviolet

UV sterilization technology has the advantages of a broad spectrum, high efficiency, and no secondary pollution. It has been proven to effectively reduce pathogenic microorganisms on the surface of meat products and extend their shelf life. It is widely used in meat preservation. In the study conducted by Söbeli et al. [66], various doses of UV were used to irradiate beef fillet, and it was observed that a higher dose (4.2 J/cm2) significantly reduced the total number of aerobic bacteria by 3.49 ± 0.67 log CFU/g. In contrast, Bryant et al. [67] investigated the effect of UV radiation on the inactivation of E. coli K12 on beef surfaces. It was shown that the microbial inactivation by UV treatment was correlated with both the duration of treatment and the distance between the sample and the light source. Generally, the longer the treatment time and the shorter the distance between UV and the sample, the higher will be the rate of microbial inactivation. The bactericidal effect of ultraviolet light varies depending on the specific type of microorganism. McLeod et al. [55] investigated the inactivation of pathogenic bacteria in chickens treated with continuous UVC. The results demonstrated that exposure to 30 J/cm2 of UVC reduced the log CFU/g counts of C. divergens, enterohemorrhagic E. coli and pseudomonas spp. by 2.8, 1.7, and 2.7 log CFU/g respectively. However, due to the poor penetration ability of UV in reaching the interior of the food, it can only reduce the survival rate of the initial bacterial count on the surface of the meat sample and cannot eliminate all microorganisms. Therefore, it is often used in combination with other sterilization technologies to enhance its effectiveness. Studies have found that the use of UV in combination with ozone [68] or other methods, such as peracetic acid and lactic acid [69], can effectively eliminate microorganisms in meat. At the same time, meat may undergo a series of oxidation processes after being exposed to a high dose of UV treatment, which can potentially affect the quality of the product. To mitigate the UV oxidation of meat lipids and proteins, hurdle technology can be employed. For example, UV can be combined with deaerator packaging, vacuum packaging, and high hydrostatic pressure [70,71]. Therefore, the development of diverse UV combination technologies is essential to ensure the quality and safety of meat in future UV applications.

Compared to fruits and vegetables, UV is used in meat and meat products to a relatively small extent. The bactericidal effect of different UV irradiation doses on meat varies. However, there is currently limited research on the relationship between dose and meat disinfection. Additionally, the application of ultraviolet in meat needs further exploration regarding appropriate wavelength, dose, and luminescence methods for different types of meat. Therefore, future work should focus on expanding basic research on UV in meat and meat products as well as researching and developing new UV technologies.

3.2.4. Cold plasma

In recent years, Cold plasma has gained widespread attention in the field of food sterilization and shows great potential in meat preservation applications. It can effectively eliminate various types of viruses, bacteria, fungi, and spores. The ability of Cold plasma to inhibit bacteria mainly depends on factors such as gas composition, airflow, electrical input, and duration of the process[72]. Kim et al. [73] used helium (10 lpm) and a mixture of helium and oxygen (10 lpm and 10 sccm) as the excitation medium to process sliced bacon under the conditions of input power at 75, 100, and 125 W, with treatment times of 60 and 90 seconds respectively. After plasma helium treatment and helium/oxygen mixed treatment, the total number of aerobic bacteria decreased by 1.89 decimals and 4.58 decimals respectively. The study also found that regardless of gas composition, the bacteria-reducing effect is enhanced with increasing input power and treatment time. In addition, low-temperature plasma has different effects on reducing bacteria in different microbial species. Choi et al. [74] used a 20 kV DC with an output voltage of 58 kHz frequency. It was found that the optimal condition for microorganism inactivation was achieved when the plasma was generated at a current of 1.5 A and the distance between the plasma electrode tip and the sample was 25 mm. After treating pork samples for 120 seconds, it was observed that E.coli O157:H7 had a stronger inactivation effect compared to Lactobacillus monocytogenes, and there were no significant changes in pork color, flavor, nutrition, or other quality indexes after treatment. Moreover, Ulbin-Figlewicz et al. [75] conducted a study on the treatment of helium and argon plasma to compare their effects on inactivating microflora on the surfaces of different types of meat. Under an operating pressure of 20 kPa, after 10 minutes of helium plasma treatment, the total number of colonies in pork decreased by 1.14 to 1.48 logarithmic cycles, while that in beef only decreased by 0.98 to 2.09 logarithmic cycles.

Currently, technology plays a significant role in the sterilization and preservation of meat and meat products; however, it is still in the early stages of basic research with some issues, such as weak penetration ability. When a large number of microorganisms gather on the food surface, resulting in uneven bactericidal effects, its effectiveness will be affected. The technology can be combined with other non-heat treatment methods to enhance its bactericidal effect. Additionally, there are challenges like high equipment investment costs. The specific influencing factors, mechanism of action, and regulatory means of this technology lack a well-established theoretical foundation for support and require further exploration in future studies to promote wider application in the meat industry.

3.2.5. HPP

The effectiveness of HPP for microbial control depends on various factors including food composition, types of microbes present, the method and magnitude of applied pressure, duration of pressure application, treatment temperature as well as the integrity of packaging [76]. Hayman et al. [77] conducted a study using HPP on different types of meat (low-fat pastrami, Strasbourg beef, export sausage, and Cajun beef), which demonstrated that treatment at 600 MPa and 20℃ for 180 seconds extended the refrigerated shelf life of ready-made meats. The amount of L. monocytogenes in the inoculated products was reduced by more than 4 log CFU/g. In order to enhance the bactericidal effect of HPP, antibacterial agents can be combined with HPP. Melhem et al. [78] investigated the combined inactivation effect of HHP on pathogenic and spoilage bacteria in meat products using different application methods (surface application, product incorporation, and active packaging) and various types of antibacterials (derived from plants, microorganisms, and animals). It was discovered that combining bacteriocin with HHP resulted in a stronger bacteriocidal effect due to its significant inhibitory impact on gram-positive bacteria like listeria. Regarding application methods, the synergistic effect of active packaging and HHP proved to be the most effective approach for pathogenic bacteria inactivation in meat products. However, the use of HPP may have an effect on the color of fresh meat, and the exact mechanism of color change is unknown. This could be attributed to the disruption of non-covalent bonds, protein denaturation, and myoglobin oxidation in fresh meat under high pressure [76,79]. Studies have shown that compared with beef or mutton treated with HPP, poultry meat has lower myoglobin content and is less affected by HPP treatment [80]. To investigate this issue, Gupta et al. [79] discovered that altering the myoglobin status can reduce the discoloration caused by HPP. Patties were prepared using different packaging methods (high oxygen - oxymyoglobin, carbon monoxide - carboxymyoglobin) or through the addition of potassium ferricyanide (metmyoglobin). It was found that carboxymyoglobin exhibited better color retention.

The application of HPP as a non-thermal physical technology in meat bacterialization has shown great potential, but it still faces some challenges. For instance, bacterial spores are highly resistant to high pressure [81], requiring higher pressures (> 1200 MPa) [28] for their inactivation. However, these higher pressure conditions may have adverse effects on food quality. Therefore, in the future, it can be used in combination with other methods to reduce bacteria and sterilize food.

3.3. Application of biological bacteria reduction technology in meat

Natural antimicrobials have long been used as food additives, primarily to extend the shelf life and maintain the quality of meat and meat products. Among natural antibacterial compounds, plant-derived antimicrobials have been extensively studied, and substances such as phenols, isoflavones, ketones, acids, terpenes (essential oils), and alkaloids commonly found in meat can limit or hinder the growth of harmful microorganisms. Different types of microorganisms in meat and meat products are inhibited by various plant extracts. Studies have shown that when thyme oil and balsam oil [82] are applied to chicken breast, balsam oil significantly restricts the growth of salmonella while thyme oil effectively inhibits the growth of E. coli. Furthermore, they are incorporated into meat in various forms, such as direct mixing, coating, layering, nanoencapsulation, and microencapsulation. This can enhance its antibacterial properties and significantly expand the application of natural antimicrobials in meat. For instance, Noshad et al. [83] blended lemon essential oil with psyllium seed slime to create a novel edible coating for beef storage and preservation, and discovered that it could extend the shelf life by 10 days. Moreover, different addition technologies may have varying inhibitory effects on microorganisms. This was demonstrated by Wang et al. [84], who found that nanoemulsions of litsea cubeba essential oil exhibited stronger inhibition against Listeria monocytogenes and Streptococcus maritimus compared to pure litsea cubeba essential oil.

Chitosan, a polycationic biopolymer commonly extracted from the exoskeletons of crustaceans such as crabs and lobsters, is frequently used as an animal-derived antimicrobial agent in meat. Darmadji et al. [85] discovered that chitosan at concentrations ranging from 0.1 % to 1% effectively inhibits the growth of spoilage bacteria and pathogenic bacteria during the storage of refrigerated meat. In addition to chitosan, lysozyme extracted from eggs and milk is also commonly employed. Furthermore, lactoferrin [86], a natural antibacterial substance found in secretions like saliva, milk, and tears of mammals, can inhibit the activity of E.coli. It is also possible to produce lactoferrin nanoparticles by electrostatically chelating gellan gum [87], thereby enhancing their antimicrobial capacity without requiring inorganic compounds. This greatly expands the utilization of lactoferrin as a natural antimicrobial agent in food.

There are various types of microorganisms, and microbial-derived agents that reduce bacteria have gradually become the focus of research. Bacteriocins are typically produced by both Gram-positive and Gram-negative bacteria [2]. Studies have shown that lactic acid bacteria isolated from buffalo milk curds produce bacteriocin with antibacterial properties, which can inhibit the growth of pathogenic microorganisms in chickens [88]. Another bacteriocin commonly used in meat is nisin, which is produced by certain strains of lactococcus lactis and has the ability to inhibit various gram-positive bacteria [89]. Arief et al. [90] achieved significant reduction in the growth of Escherichia coli in mutton intestines by adding probiotic lactobacillus plantarum IIA-2C12, which also preserved the color of meat products. However, this addition also resulted in changes to the taste of the meat and decreased acceptability to some extent. Therefore, investigating the impact of microbial preservatives on the flavor of meat products is an important area for future research. Araujo et al. [91] also investigated the bacterioreducing effects of a combination of garlic essential oil (GO), allyl isothiocyanate (AITC), and nisin (NI) on fresh intestines. The combination of 20 mg/kg NI + 125 μL/kg GO + 62.5 μL/kg AITC or 20 mg/kg NI + 62.5 μL/kg GO + 125 μL/kg AITC was found to have a significant inhibitory effect on E. coli O157H7 and spoilage lactic acid bacteria, which was better than that of single applications, without affecting the sensory acceptability of fresh intestine.

  • In the revised manuscript, the sentences about the comparison among the technologies and the summary of the technologies have been supplemented in line 77-199

Table 1.The classification and characteristics of bacteria-reducing technology

Classification

Advantage

Disadvantage

Chemical bacteria reduction technology

SAEW

Efficient sterilization, convenient manufacturing, low cost, wide application, safety and environmental protection

Effect instability

Organic acids

Low cost, green, efficient sterilization

Unstable and easy to decompose

Ozone

Efficient sterilization, no secondary pollution

High equipment cost and poor stability

Non-thermal physical bacteria reduction technology

Ultrasound

Green safety, wide applicability, convenient and fast

Limited penetration, uneven sterilization

Radiation

Efficient sterilization, cold treatment, no residue, strong controllability, wide application, environmental protection and energy saving

High cost, high equipment and technical requirements

UV

Environmentally friendly, no residue

High energy consumption, safety risks, penetration limitations, and environmental impacts

Cold plasma

Mild, efficient, no residue

Complex technology, poor stability, technical maturity

HPP

High sterilization efficiency, low energy consumption, green and safe

High equipment cost and application limitation

Biological bacteria reduction technology

Plant-derived natural antimicrobials

Environmentally friendly, green and safe

Obviously seasonal and regional, the effect is affected by the separation and extraction process

Animal-derived natural antimicrobials

Natural origin, broad spectrum, safety, biocompatibility and ease of application

Drug resistance and antimicrobial function are limited

Microbial-derived natural antimicrobials

Wide range of sources, high security

Effect instability

2.1 Chemical bacteria reduction technology

Slightly acidic electrolyzed water (SAEW) is electrolyzed water with a certain available chlorine content (ACC) by applying a direct current voltage to an electrolytic cell without diaphragm [13]. Its pH is generally 5.0 ~ 6.5 and oxidation reduction potential (ORP) ≧600 mV. It has been found that the HClO in SAEW, which exhibits a highly potent bactericidal effect, can result in a microbial lethality rate of over 90 % [14]. SAEW attacks the cell wall, cell membrane and intracellular components of microbial cells to achieve the purpose of sterilization, effectively reducing the common foodborne pathogens in food. Moreover, due to the short action time of SAEW, it will not cause bacterial resistance to develop. It possesses characteristics such as efficient sterilization, convenient manufacturing, low cost, wide application, safety and environmental friendliness. Therefore, it is a chemical technology for reducing bacteria with broad prospects for development. However, strict control of the pH range of SAEW is necessary to avoid compromising its bactericidal effect. Additionally, long-term use may result in corrosion and damage to metal equipment surfaces.

Organic acids, as food-grade antibacterial agents, are a class of organic compounds that contain carboxylic (- COOH) functional groups. They can inhibit the growth of microorganisms by reducing the pH value of food and play a bactericidal role. Organic acids commonly used for reducing bacteria in food include lactic acid, propionic acid, sorbic acid, acetic acid, citric acid, and tartaric acid. These acids are primarily produced through microbial fermentation engineering or naturally occurring processes. They are widely employed in food preservation due to their low cost, natural coloration, high bactericidal efficiency, and international recognition as safe (GRAS) bactericidal agents [15]. In addition to their role as bacterial-reducing agents in food, organic acids are also listed in FDA regulations for various technical applications such as acidity regulators, antioxidants, flavor enhancers, and pH adjusters [16]. It should be noted that relatively high concentrations of certain organic acids may affect the taste and flavor of the food, as well as potentially persist in it, thus requiring reasonable control during usage.

Ozone (O3) is a light blue gas with a distinct odor, which is produced by an ozone generator through high voltage discharge or ultraviolet radiation alone or in combination with nitrogen oxides (NOx) [17]. Ozone is commonly used in both gaseous and aqueous forms in the food industry. It acts on cell walls, cell membranes, intracellular enzymes, genetic material, fungal spore shells, or viral capsids to inactivate a variety of microorganisms through ozone itself or its breakdown products such as hydroxyl radicals [18]. It is also easy to decompose into HO- and O2 in water, which has a strong disinfecting and sterilising ability[19] Ozone automatically and rapidly breaks down in the air to produce oxygen, leaving no residue in food. Furthermore, ozone can quickly diffuse throughout a space, ensuring sterilization without any missed areas. Due to its high sterilization efficiency, and lack of pollution [20], ozone is widely used in food preservation and other fields. However, ozone treatment requires specialized equipment and technology. The operation process is relatively complex, and high concentrations of ozone are harmful to the human body. Therefore, strict control of concentration and contact time is necessary when using it.

2.2 Non-thermal physical bacteria reduction technology

Ultrasound is a longitudinal mechanical wave beyond the range of human hearing, with a frequency generally between 20 kHz and 1 GHz [21]. It can travel through air, liquids, and solids and has the capability to kill certain microorganisms in food. In the food industry, ultrasound at a lower frequency (20 ~ 100 KHz) is usually used to deactivate microorganisms. The bactericidal mechanism of ultrasonic waves is mainly attributed to the cavitation effect [22]. Under the influence of ultrasonic waves, cavitation bubbles oscillate, grow, and burst, resulting in instantaneous high temperature and pressure. The rapid alternation of temperature and pressure directly damages the cell wall or membrane of microorganisms, promotes water molecule decomposition, and triggers free radical reactions. These generated free radicals possess strong oxidizing properties that damage the DNA and enzymes of microbial cells, leading to microbial deactivation [23]. The technology possesses characteristics such as low cost, green safety, wide applicability, and convenience. However, the penetration ability of ultrasonic waves is limited; therefore, it is commonly combined with other bacterial-reducing technologies in practical applications.

Irradiation sterilisation technology utilizes ionising radiation sources of various wavelengths, such as electron beams, γ-rays or X-rays to radiate food [24], reducing or removing most of the harmful microorganisms in food. This technology can minimize the decline in food quality, extend shelf life, and is widely used in the field of food processing and preservation. After irradiation, microorganisms in food (such as bacteria, yeast, and mold) absorb radiation energy which breaks chemical bonds and changes cytochemical composition to achieve sterilization [24]. In general, to ensure the safety of processed meat, the maximum absorbed dose of irradiated food should not exceed 10 kGy according to the standards established by the Codex Alimentarius Commission (CAC) [25]. However, different countries have varying standards for using irradiation in food. In the food industry, it is also necessary to determine the irradiation dose based on different types of food in order to effectively kill harmful microorganisms while maintaining food quality. The high cost of setting up and maintaining irradiation facilities may limit their use in certain areas. Furthermore, the use of irradiation necessitates professional equipment and skilled personnel. Factors such as cost, technology, and regulations need to be comprehensively considered. However, it is a highly efficient method for sterilization, cold treatment with no residue, strong controllability, and environmentally friendly food treatment that saves energy. With the advancement of technology and the improvement of public awareness, irradiation technology is expected to be applied in more fields.

Ultraviolet (UV), as a non-ionizing radiation, is approved by the FDA as a cold sterilization technology for reducing the microbial content of food. The wavelength of the UV emission spectrum ranges from 100 nm to 400 nm, which is longer than that of X-rays but shorter than that of visible light. According to the wavelength range, it can be divided into long-wave ultraviolet (UVA, 315~400 nm), medium-wave ultraviolet (UVB, 280~315 nm), and short-wave ultraviolet (UVC, 100~280 nm). Among them, UVC is the most commonly used ultraviolet range in food processing and has high single-particle energy. It can kill bacteria, fungi, viruses, and other microorganisms by altering the permeability of microbial cell membranes and damaging microbial DNA or RNA [26]. In the process of sterilization, UV does not need to add any chemical substances, nor does it leave any harmful residues in the food [27]. It is considered a broad-spectrum, efficient, environmentally friendly, and residue-free technology for reducing bacteria. The penetration of ultraviolet light, however, is weak, and it can only sterilize the directly exposed surface. It belongs to ionizing radiation and requires attention to safety protection measures when used.

Cold plasma is a type of electroneutral ionized gas, which is composed of electrons, positive and negative ions, reactive oxygen species, excited or non-excited gas molecules and photons. It is under the action of heat, electric field and microwave, so that some of the atoms or molecules, are ionised to form substances based on reactive oxygen, reactive nitrogen and ultra violet (UV) photons. These substances interact with the food surface and cause microbial death. The production process of plasma involves high-energy electrons, and strict safety measures are required during operation to avoid injury to the operator. Compared with traditional heat treatment or chemical treatment technology, the technical maturity and market application of cold plasma sterilization technology is still developing and needs further verification and improvement. However, cold plasma sterilization technology has the characteristics of being mild, highly efficient, and residue-free, which will play an important role in optimizing the meat preservation process.

High-pressure processing (HPP) involves placing food in a fluid medium after soft packaging at room temperature or low temperature and subjecting it to pressure ranging from 100 to 1000 MPa for a certain period of time. This allows the pressure to be uniformly transmitted to the food [28], resulting in lethal effects on microorganisms. The purpose of sterilization is achieved by affecting gene expression, destroying the cytoplasmic membrane, denaturing proteins, and inactivating metabolic enzymes. Since covalent bonds are generally not broken by HPP, the aroma compounds of vitamins or foods are usually unaffected [29]. In contrast, large molecules such as carbohydrates and proteins undergo structural changes under stress. Despite the high equipment cost of HPP in the production process, especially in large-scale applications, which results in a significant investment that may affect the texture of food, it offers advantages such as short processing time, low energy consumption, uniform pressure effect, and green safety. Additionally, it can extend the shelf life of food and maintain its nutritional value [30]. Therefore, HPP has a promising market application prospect.

Q2: did authors used ai for manuscript writting?

A2: Thanks for your review carefully. The manuscript was not written by AI. The manuscript was written on the basis of our previous research and a large number of literatures searched from Web of Science, Science Direct, Scopus, PubMed and other databases.

Q3: Authors need to include 2 paragraphs minimum for each sterilization technique, summarizing published reports.

A3: Thanks for your reminding. In the revised manuscript, the summaries have been supplemented in line 255-567 as follow:

3.1. Application of chemical bacteria reduction technology in meat

3.1.1. SAEW

The bactericidal effect of SAEW varies slightly depending on the treatment method used, such as soaking, atomization, spraying, washing, etc. Among these methods, soaking treatment yields the best bactericidal effect. Rahman et al. [41] soaked fresh chicken breast with SAEW, which significantly reduced the number of natural flora and pathogenic bacteria, and slowed down the rate of spoilage during refrigeration. Similarly, Sheng et al. [42] discovered that soaking beef in SAEW was effective in killing microorganisms and delaying the deterioration of refrigerated beef quality. Furthermore, the bactericidal effect of SAEW varies depending on the duration and concentration of treatment. Ye et al. [43] investigated the effects of different treatment times and concentrations of SAEW on raw frozen shrimp and found that treating with SAEW for 5 minutes had a better bactericidal effect than treating with SAEW for 3 minutes. Additionally, under the same treatment time, the bactericidal effect of 29 mg/L of SAEW was significantly enhanced compared to that of 16 mg/L of SAEW. However, SAEW has limited ability to maintain oxidative stability during meat storage. Therefore, combining SAEW with antioxidants can help enhance its oxidative stability. However, further research is needed. In recent years, SAEW ice has been studied and applied for the preservation of meat and aquatic products [44]. The storage and preservation of aquatic products with ice containing bactericidal substances can not only inhibit the reproduction of microorganisms on the surface of the products but also extend their shelf life.

Currently, although SAEW technology has been widely utilized in the domestic and international food industry, there is still a lack of systematic research and evaluation on the sterilization principle of SAEW, influencing factors, and its impact on the oxidation of different types of meat. This deficiency also hinders the promotion and application of SAEW in meat processing and industrialization. Maximizing the sterilization and preservation effect of SAEW, as well as exploring its combination with other technologies, remains the focus of future research.

3.1.2. Organic acids

The bacterial reduction effect of organic acids is closely related to their species, concentration, pH, volume, the type of target microorganisms, and temperature of the carcass surface. Organic acids are often sprayed on meat to reduce bacteria. However, soaking the meat in organic acids can lead to cross-contamination and significant loss of soluble substances, which affects the effectiveness of bacteria reduction. Some scholars have explored the bacteria-reducing effect of nine different organic acids (including lactic acid, acetic acid, citric acid, peracetic acid, etc.) on cattle carcasses and found that lactic acid is the most effective measure for reducing bacteria. It can effectively reduce the total number of colonies and coliform bacteria [45]. In addition, lactic acid is the most commonly used organic acid in the meat industry to reduce bacteria in products due to its effectiveness and cost. Furthermore, the European Union permits the spraying of 2% to 5% lactic acid on carcasses as a means of reducing bacteria [46]. Ransom et al. [47]demonstrated that applying 2% lactic acid can decrease E. coli O157:H7 on cattle carcass surfaces by 1.6 Log (CFU/g). Manzoor et al. [48] also compared the effects of different concentrations of lactic acid spray on the microbial and sensory characteristics of buffalo meat and found that both could significantly reduce the number of microorganisms, while a concentration of lactic acid above 6% would adversely affect the color of the meat. When a higher temperature (above 55℃) is combined with a lower concentration (2 %) of lactic acid spray, the bactericidal effect is enhanced. However, the stability of organic acids is compromised under high temperature conditions, leading to changes in meat's sensory properties and the emergence of acid-resistant pathogens. Additionally, this may also result in corrosion to processing equipment [45]. The study found that the combination of two or more organic acids is more effective than a single organic acid, which can enhance the bactericidal effect and overall food quality. Surve et al. [49] found that combining acetic acid with lactic acid or propionic acid can significantly enhance the antimicrobial properties of buffalo meat without affecting its color and flavor, effectively extending its refrigerated shelf life.

In the future, new organic acids or derivatives of organic acids could be developed to address the issues of heat resistance and broad-spectrum antimicrobial properties associated with existing organic acids. Despite some limitations in food sterilization, organic acids have broad application prospects as a natural and environmentally friendly sterilization technology due to increasing consumer demand for natural and safe foods.

3.1.3. Ozone

The application of ozone in meat can be divided into two forms: gaseous state and aqueous solution[12]. Its safety and effectiveness have been proven for killing a variety of foodborne pathogens in food, such as Escherichia coli, Salmonella, and Listeria monocytogenes [50]. Cárdenas et al. [51] analyzed the antibacterial effect of gaseous ozone on chilled fresh beef and found that an ozone concentration of 141.12 mg/m3 could reduce the total number of E. coli and inoculated microorganisms, resulting in complete inactivation of certain microorganisms and maintaining meat freshness. The bactericidal effect of ozone is influenced by factors such as concentration, temperature, contact time, and food characteristics. Stivarius et al. [52] investigated the bactericidal effect of a 1% ozone aqueous solution and found that a 7-minute ozone treatment had a better bactericidal effect than a 5-minute treatment. Additionally, the researchers compared high ozone dose (1000 ppm) with low ozone dose (100 ppm) and discovered that although the high ozone dose can slow down microbial activity on the surface of pork and reduce physiological activity, it is not sufficient to have a fatal effect on microbial life in meat. This may be attributed to the longer incubation time (46 and 49 hours) of samples under non-sterile conditions after ozone treatment [53]. However, ozone is also a potent oxidizing agent that can impact the sensory, physical, and chemical properties of meat and meat products. Ayranci et al. [54] discovered that ozone treatment (1 x 10-2 kg m-3, 8 hours) had detrimental effects on the physicochemical properties and color of turkey breast. The high oxidation potential of ozone may be the primary reason for lipid and protein oxidation in meat. Additionally, it was observed that the connective tissue membrane with a high protein content in turkey meat underwent denaturation after ozone treatment, which could explain the lighter coloration of turkey meat.

Ozone sterilization systems require complex equipment and precise control, making them costly to operate and maintain. Furthermore, high concentrations of ozone can be toxic to the human body, so it is necessary to strictly regulate the concentration and contact time during actual operation in order to ensure food safety and protect operators.

3.2. Application of non-thermal physical bacteria reduction technology in meat

3.2.1. Ultrasound

In the food industry, high-power ultrasonic waves with a frequency of 20-100 kHz are generally used to inactivate microorganisms [55]. The sterilization effect of ultrasonication is mainly influenced by the ultrasonic medium, frequency, action time, and microbial species [56]. Huu, et al.[57] used ultrasonic treatments with a frequency of 40 kHz and a power density of 0.092 W/mL for E. coli O157:H7 or L. innocua. The initial bacterial content was 1 × 106 CFU/mL, and the bacterial content remained at the initial level after both 30 min and 45 min of ultrasonic treatment. The results showed that there was no significant influence on either bacteria in the product during the entire period of ultrasonic treatment. It has been found that the combination of ultrasonic and heat treatment can accelerate the sterilization speed of food. Morild et al. [58] evaluated the effect of pressurized steam combined with high-power ultrasound on pathogen inactivation on pig skin and pork surface. The study examined the inactivation of Salmonella typhimurium, Salmonella Derby, Salmonella infantile, Yersinia enterocolitica, and a non-pathogenic E. coli. After 4 seconds of ultrasound treatment, the total number of colonies decreased by 3.3 log CFU/cm2. Musavian et al. [59] also found that steam treatment and ultrasonic treatment of chicken carcasses on the processing line could significantly reduce the number of campylobacter on contaminated poultry. By using steam and ultrasound immediately after slaughter, the total count of viable bacteria was reduced by approximately 3 log CFU/cm2. However, prolonged exposure of food to high temperatures at ultrasonic wavelengths leads to a reduction in its functional properties, sensory properties, and nutritional value [60]. Therefore, it can be combined with other technologies that reduce bacteria to enhance its bactericidal effect. Kordowska-Wiater et al. [61] investigated the isolation of Salmonella, Escherichia coli, Proteus, and Pseudomonas fluorescence from the surface of chicken skin after being treated with ultrasound (at a frequency of 40 kHz, intensity of 2.5 W cm2, and duration of either 3 or 6 minutes) in water and a 1% lactic acid solution. It was observed that treating the chicken skin with ultrasound alone in a lactic acid solution for 3 minutes resulted in a reduction of 1.0 log CFU/cm2 in the number of these microorganisms. Furthermore, extending the treatment time to 6 minutes led to a decrease exceeding 1.0 log CFU/cm2 in the number of microorganisms present in the water sample.

Ultrasound have varying effects on the inactivation of different microorganisms (bacteria, viruses, fungi, and mycotoxins) as well as food substrates. The frequency, intensity, and treatment time of ultrasonic waves need to be optimized for different types of food. Several studies have shown that while ultrasound can reduce the number of bacteria produced by meat, using two or more bacteria-reducing technologies is more effective than relying solely on bacterial reduction [21]. Therefore, it is rarely used alone in actual meat production processes for reducing bacteria and usually works in conjunction with other bacteria-reducing technologies.

3.2.2. Irradiation

Irradiation technology has been approved by the Food and Drug Administration (FDA) for use in food processing. Food irradiation is a form of "cold treatment" that effectively kills bacteria without significantly increasing the internal temperature or causing nutrient loss in the food [47]. Electron beam radiation (EB) and X-ray radiation (XR) are more acceptable to consumers than gamma rays (GR) because they do not involve radioactive isotopes [48]. However, Park et al. [62] found that GR irradiation may be more effective in reducing bacterial populations than EB irradiation. Additionally, doses of 5-10 kGy for GR and EB irradiation respectively had no adverse effects on the lipid oxidation and sensory characteristics (color, chewability, and taste) of beef sausage patties. The technical advantages of high-energy X-rays include better power utilization, dose uniformity, and shorter irradiation times. Consequently, the use of X-ray irradiation results in higher productivity and lower processing costs. Yim et al. [63] used X-ray irradiation to treat beef samples, and as the irradiation dose increased, there was a significant decrease in the total number of aerobic bacteria present. No bacterial growth was observed in samples treated with a dose of 10 kGy. The combined treatment of irradiation and natural antimicrobial agents in raw meat materials can enhance the degree of inhibition against food-borne pathogens and extend the shelf life of meat and meat products. Hu et al. [64] combined chitosan-eugenol with irradiation to significantly reduce the number of Staphylococcus aureus and Salmonella in fresh pork, as well as delay fat oxidation during processing.

However, although the meat and its products treated with low-dose radiation will not be contaminated by radioactivity and play a good role in inhibiting the growth of pathogenic microorganisms in meat, it is important to select the appropriate radiation for each type of food during use. Additionally, parameters such as radiation dose and time should be well controlled to minimize adverse effects on the quality of meat flavor, color, nutrition, and moisture [65].

3.2.3. Ultraviolet

UV sterilization technology has the advantages of a broad spectrum, high efficiency, and no secondary pollution. It has been proven to effectively reduce pathogenic microorganisms on the surface of meat products and extend their shelf life. It is widely used in meat preservation. In the study conducted by Söbeli et al. [66], various doses of UV were used to irradiate beef fillet, and it was observed that a higher dose (4.2 J/cm2) significantly reduced the total number of aerobic bacteria by 3.49 ± 0.67 log CFU/g. In contrast, Bryant et al. [67] investigated the effect of UV radiation on the inactivation of E. coli K12 on beef surfaces. It was shown that the microbial inactivation by UV treatment was correlated with both the duration of treatment and the distance between the sample and the light source. Generally, the longer the treatment time and the shorter the distance between UV and the sample, the higher will be the rate of microbial inactivation. The bactericidal effect of ultraviolet light varies depending on the specific type of microorganism. McLeod et al. [55] investigated the inactivation of pathogenic bacteria in chickens treated with continuous UVC. The results demonstrated that exposure to 30 J/cm2 of UVC reduced the log CFU/g counts of C. divergens, enterohemorrhagic E. coli and pseudomonas spp. by 2.8, 1.7, and 2.7 log CFU/g respectively. However, due to the poor penetration ability of UV in reaching the interior of the food, it can only reduce the survival rate of the initial bacterial count on the surface of the meat sample and cannot eliminate all microorganisms. Therefore, it is often used in combination with other sterilization technologies to enhance its effectiveness. Studies have found that the use of UV in combination with ozone [68] or other methods, such as peracetic acid and lactic acid [69], can effectively eliminate microorganisms in meat. At the same time, meat may undergo a series of oxidation processes after being exposed to a high dose of UV treatment, which can potentially affect the quality of the product. To mitigate the UV oxidation of meat lipids and proteins, hurdle technology can be employed. For example, UV can be combined with deaerator packaging, vacuum packaging, and high hydrostatic pressure [70,71]. Therefore, the development of diverse UV combination technologies is essential to ensure the quality and safety of meat in future UV applications.

Compared to fruits and vegetables, UV is used in meat and meat products to a relatively small extent. The bactericidal effect of different UV irradiation doses on meat varies. However, there is currently limited research on the relationship between dose and meat disinfection. Additionally, the application of ultraviolet in meat needs further exploration regarding appropriate wavelength, dose, and luminescence methods for different types of meat. Therefore, future work should focus on expanding basic research on UV in meat and meat products as well as researching and developing new UV technologies.

3.2.4. Cold plasma

In recent years, Cold plasma has gained widespread attention in the field of food sterilization and shows great potential in meat preservation applications. It can effectively eliminate various types of viruses, bacteria, fungi, and spores. The ability of Cold plasma to inhibit bacteria mainly depends on factors such as gas composition, airflow, electrical input, and duration of the process[72]. Kim et al. [73] used helium (10 lpm) and a mixture of helium and oxygen (10 lpm and 10 sccm) as the excitation medium to process sliced bacon under the conditions of input power at 75, 100, and 125 W, with treatment times of 60 and 90 seconds respectively. After plasma helium treatment and helium/oxygen mixed treatment, the total number of aerobic bacteria decreased by 1.89 decimals and 4.58 decimals respectively. The study also found that regardless of gas composition, the bacteria-reducing effect is enhanced with increasing input power and treatment time. In addition, low-temperature plasma has different effects on reducing bacteria in different microbial species. Choi et al. [74] used a 20 kV DC with an output voltage of 58 kHz frequency. It was found that the optimal condition for microorganism inactivation was achieved when the plasma was generated at a current of 1.5 A and the distance between the plasma electrode tip and the sample was 25 mm. After treating pork samples for 120 seconds, it was observed that E.coli O157:H7 had a stronger inactivation effect compared to Lactobacillus monocytogenes, and there were no significant changes in pork color, flavor, nutrition, or other quality indexes after treatment. Moreover, Ulbin-Figlewicz et al. [75] conducted a study on the treatment of helium and argon plasma to compare their effects on inactivating microflora on the surfaces of different types of meat. Under an operating pressure of 20 kPa, after 10 minutes of helium plasma treatment, the total number of colonies in pork decreased by 1.14 to 1.48 logarithmic cycles, while that in beef only decreased by 0.98 to 2.09 logarithmic cycles.

Currently, technology plays a significant role in the sterilization and preservation of meat and meat products; however, it is still in the early stages of basic research with some issues, such as weak penetration ability. When a large number of microorganisms gather on the food surface, resulting in uneven bactericidal effects, its effectiveness will be affected. The technology can be combined with other non-heat treatment methods to enhance its bactericidal effect. Additionally, there are challenges like high equipment investment costs. The specific influencing factors, mechanism of action, and regulatory means of this technology lack a well-established theoretical foundation for support and require further exploration in future studies to promote wider application in the meat industry.

3.2.5. HPP

The effectiveness of HPP for microbial control depends on various factors including food composition, types of microbes present, the method and magnitude of applied pressure, duration of pressure application, treatment temperature as well as the integrity of packaging [76]. Hayman et al. [77] conducted a study using HPP on different types of meat (low-fat pastrami, Strasbourg beef, export sausage, and Cajun beef), which demonstrated that treatment at 600 MPa and 20℃ for 180 seconds extended the refrigerated shelf life of ready-made meats. The amount of L. monocytogenes in the inoculated products was reduced by more than 4 log CFU/g. In order to enhance the bactericidal effect of HPP, antibacterial agents can be combined with HPP. Melhem et al. [78] investigated the combined inactivation effect of HHP on pathogenic and spoilage bacteria in meat products using different application methods (surface application, product incorporation, and active packaging) and various types of antibacterials (derived from plants, microorganisms, and animals). It was discovered that combining bacteriocin with HHP resulted in a stronger bacteriocidal effect due to its significant inhibitory impact on gram-positive bacteria like listeria. Regarding application methods, the synergistic effect of active packaging and HHP proved to be the most effective approach for pathogenic bacteria inactivation in meat products. However, the use of HPP may have an effect on the color of fresh meat, and the exact mechanism of color change is unknown. This could be attributed to the disruption of non-covalent bonds, protein denaturation, and myoglobin oxidation in fresh meat under high pressure [76,79]. Studies have shown that compared with beef or mutton treated with HPP, poultry meat has lower myoglobin content and is less affected by HPP treatment [80]. To investigate this issue, Gupta et al. [79] discovered that altering the myoglobin status can reduce the discoloration caused by HPP. Patties were prepared using different packaging methods (high oxygen - oxymyoglobin, carbon monoxide - carboxymyoglobin) or through the addition of potassium ferricyanide (metmyoglobin). It was found that carboxymyoglobin exhibited better color retention.

The application of HPP as a non-thermal physical technology in meat bacterialization has shown great potential, but it still faces some challenges. For instance, bacterial spores are highly resistant to high pressure [81], requiring higher pressures (> 1200 MPa) [28] for their inactivation. However, these higher pressure conditions may have adverse effects on food quality. Therefore, in the future, it can be used in combination with other methods to reduce bacteria and sterilize food.

3.3. Application of biological bacteria reduction technology in meat

Natural antimicrobials have long been used as food additives, primarily to extend the shelf life and maintain the quality of meat and meat products. Among natural antibacterial compounds, plant-derived antimicrobials have been extensively studied, and substances such as phenols, isoflavones, ketones, acids, terpenes (essential oils), and alkaloids commonly found in meat can limit or hinder the growth of harmful microorganisms. Different types of microorganisms in meat and meat products are inhibited by various plant extracts. Studies have shown that when thyme oil and balsam oil [82] are applied to chicken breast, balsam oil significantly restricts the growth of salmonella while thyme oil effectively inhibits the growth of E. coli. Furthermore, they are incorporated into meat in various forms, such as direct mixing, coating, layering, nanoencapsulation, and microencapsulation. This can enhance its antibacterial properties and significantly expand the application of natural antimicrobials in meat. For instance, Noshad et al. [83] blended lemon essential oil with psyllium seed slime to create a novel edible coating for beef storage and preservation, and discovered that it could extend the shelf life by 10 days. Moreover, different addition technologies may have varying inhibitory effects on microorganisms. This was demonstrated by Wang et al. [84], who found that nanoemulsions of litsea cubeba essential oil exhibited stronger inhibition against Listeria monocytogenes and Streptococcus maritimus compared to pure litsea cubeba essential oil.

Chitosan, a polycationic biopolymer commonly extracted from the exoskeletons of crustaceans such as crabs and lobsters, is frequently used as an animal-derived antimicrobial agent in meat. Darmadji et al. [85] discovered that chitosan at concentrations ranging from 0.1 % to 1 % effectively inhibits the growth of spoilage bacteria and pathogenic bacteria during the storage of refrigerated meat. In addition to chitosan, lysozyme extracted from eggs and milk is also commonly employed. Furthermore, lactoferrin [86], a natural antibacterial substance found in secretions like saliva, milk, and tears of mammals, can inhibit the activity of E.coli. It is also possible to produce lactoferrin nanoparticles by electrostatically chelating gellan gum [87], thereby enhancing their antimicrobial capacity without requiring inorganic compounds. This greatly expands the utilization of lactoferrin as a natural antimicrobial agent in food.

There are various types of microorganisms, and microbial-derived agents that reduce bacteria have gradually become the focus of research. Bacteriocins are typically produced by both Gram-positive and Gram-negative bacteria [2]. Studies have shown that lactic acid bacteria isolated from buffalo milk curds produce bacteriocin with antibacterial properties, which can inhibit the growth of pathogenic microorganisms in chickens [88]. Another bacteriocin commonly used in meat is nisin, which is produced by certain strains of lactococcus lactis and has the ability to inhibit various gram-positive bacteria [89]. Arief et al. [90] achieved significant reduction in the growth of Escherichia coli in mutton intestines by adding probiotic lactobacillus plantarum IIA-2C12, which also preserved the color of meat products. However, this addition also resulted in changes to the taste of the meat and decreased acceptability to some extent. Therefore, investigating the impact of microbial preservatives on the flavor of meat products is an important area for future research. Araujo et al. [91] also investigated the bacterioreducing effects of a combination of garlic essential oil (GO), allyl isothiocyanate (AITC), and nisin (NI) on fresh intestines. The combination of 20 mg/kg NI + 125 μL/kg GO + 62.5 μL/kg AITC or 20 mg/kg NI + 62.5 μL/kg GO + 125 μL/kg AITC was found to have a significant inhibitory effect on E. coli O157H7 and spoilage lactic acid bacteria, which was better than that of single applications, without affecting the sensory acceptability of fresh intestine.

However, in practical application, the development and utilization of natural antimicrobial agents are still imperfect, with problems such as instability, insufficient extraction of available active substances, and limited effectiveness of a single antimicrobial agent, which may prevent achieving the expected results. To better utilize natural antimicrobials, new composite antimicrobials can be synthesized by optimizing their extraction and purification methods and utilizing new technologies [84] to enhance their stability and antibacterial effects. This will facilitate their improved application in the field of meat preservation.

Q4: Also comparison between sterilization yechniques, in terms of advantages/disadvantages is missing.

A4:Thanks for your suggestion. In the revised manuscript, the comparison of the advantages and disadvantages among these technologies have been supplemented, and the comparison information has been summerized in Table 1., just as the answer of A1 (2)

Q5: Content of review is missing.

A5: Thanks for your review carefully. In the revised manuscript, the research status and purpose of the review, the comparison of the advantages and disadvantages among these technologies, the parameters of the technologies, and the introduction of the hurdle technology have been supplemented as follow:

  • The sentences about the research status and purpose of the introduction have been supplemented in line 55-64 as follow:

Previous studies have extensively examined the application of single bacteriological reduction technology in food [10–12]. However, there has been a lack of research on the application of different bacteriological reduction technologies in meat preservation in re-cent years. This paper reviews several commonly used pretreatment technologies for re-ducing bacteria, analyzes their technical principles and mechanisms for inhibiting bacte-rial growth, and summarizes their application in meat preservation. Furthermore, it dis-cusses the advantages and disadvantages of these bacteria-reducing technologies as well as their future development trends. The aim is to provide a scientific theoretical reference for better understanding and application of bacteria-reducing technologies in meat preservation.

  • The comparison of the advantages and disadvantages among these technologies have been supplemented, and the comparison information has been summerized in Table 1., just as the answer of A1 (2).
  • The sentences about the parameters of each technology have been supplemented just as the answer of A1 (1).
  • The sentences about the hurdle technology have been supplemented in line 46-52, 568-579 as follow:

The hurdle technology is a method for ensuring the safety of food products by controlling or destroying microbial spoilage or food pathogens through the combination of a series of preservative parameters (hurdles) imposing physiological impact on microbial cells [4,5]. Many researchers utilize physical sterilization technology, chemical bactericidal technology, biological bactericidal technology, and low temperature preservation technology as a combination of defense factors to achieve the preservation effect of “1+1 > 2” through fence technology [6–9].

3.4. Application of hurdle technology in Meat

The above bacteriological reduction technologie can be used to improve the safety and shelf life of food ingredients. However, the antibacterial effects of some of these technologies are not obvious when used alone and adversely affect the organoleptic properties of foods and reduce consumer acceptability. Owing to these facts, the hurdle concept (generally known as combined methods, combination preservation, combined processes, barrier technology or combination techniques) has become a promising technology that simultaneously reduces losses of nutritional and sensory quality and improves food safety [41]. The hurdle technology is a method for ensuring the safety of food products by controlling or destroying microbial spoilage or food pathogens through the combination of a series of preservative parameters (hurdles) imposing physiological impact on microbial cells. Early studies have shown that the preservation period of foods with low initial bacterial counts can be extended 1 ~ 2 times longer than those with high initial bacterial counts. If the initial amount of bacteria inside the food is low, only a few defense factors are needed for effective bacterial inhibition; conversely, if poor sanitary conditions cause a high initial amount of bacteria, it is necessary to increase the defense factor or enhance its strength.

Reviewer 2 Report

Comments and Suggestions for Authors

The authors have meticulously compiled the various pretreatment methods for meat preservation. I do not have any major comments. The only comment that I have is to improve the presentation of the concept in the Figure. I also recommend recreating this figure completely to improve its quality, as it seems to be pixelated in its current format.

Author Response

Response to Reviewer #2

Thank you very much for the recognition of this study. We would like to express our most sincere appreciation for your careful review of this manuscript. The valuable suggestions you had proposed were beneficial to improve the quality of this manuscript. The modifications have been performed carefully according to your suggestions as follow:

Q1: The authors have meticulously compiled the various pretreatment methods for meat preservation. I do not have any major comments. The only comment that I have is to improve the presentation of the concept in the Figure. I also recommend recreating this figure completely to improve its quality, as it seems to be pixelated in its current format.

A1: Thanks for your suggestion. In the revised manuscript, the Figure 1 has been redrawn, the mechanism of bacteria-reducing technology of meat has been also supplemented in Table 2 in line 250 as follow:

Figure 1. Mechanism of bacteria reducing technology.

Table 2. Mechanism of bacteriostasis in meat reducing technology

Bacteriological reduction technology

Categorisation

Mode of action

Antibacterial mechanism

Chemical bacteria reduction technology

SAEW

HCIO

CIO-

Reactive oxygen species

The cell membrane is damaged by SAEW, causing rapid leakage of K+ and an increase in membrane permeability. This causes HClO and ClO- to enter the cell, resulting in the following consequences:

(1)  It causes the breakage of DNA base, which inhibits the protein biosynthesis;

(2)  It causes the breakage of protein structure;

(3)  It causes changes in electron flow, inhibiting microbial energy metabolism and ATP production [32].

Organic acids

RCOOH

COOH-

RCOOH enters into the cell, leading to the following consequences:

(1)  RCOOH ionized by RCOOH causes the breakage of DNA base and DNA strand, which inhibits the protein biosynthesis;

(2)  The H+ ionized by RCOOH reduce the pH value of the cells, and H+ is excreted out of the cell through energy released by the transformation from ATP into ADP. The consumption of a large amount of energy inhibits the reproduction of bacteria.

Ozone

O3

O3 increases the permeability of the cell membrane and destroys lipoproteins and lipopolysaccharides, resulting in the following results after entering the cell:

(1)  It causes the breakage of enzyme structure;

(2)  It disrupts the structures of DNA and RNA, interfering with protein synthesis [19,33].

Non-thermal physical bacteria reduction technology

Ultrasound

Cavitation effect

Reactive oxygen species

Microbial cells experience violent oscillations that disrupt the permeability of cell membranes and release reactive oxygen species enter the cell, which results following consequences:

(1)  The water molecules in the liquid medium decompose into reactive oxygen species, causing reduction reactions and secondary oxidation, which results in changes to the structure of the solute [34];

(2)  It causes the breakage of enzyme structure [35].

Radiation

Electron rays

γ-rays

X-rays

(1)  Electron rays damage the inner cell membrane, leading to disruption of the enzyme system and impairment of functions;

(2)  It causes the breakage of DNA bases of microorganisms [36].

UV

UV

It destroys the DNA base, inhibits DNA transcription, replication and cell division, and inhibits protein synthesis by altering or destroying the structure of DNA or RNA molecules [1].

Cold plasma

Active substances

Reactive oxygen species

Electrically charged particles

Reactive Oxygen Species (ROS), Reactive Nitrogen Species (RNS), and charged particles destroy bacterial cells and then enter the cell interior, resulting in the following consequences:

(1)          It causes the destruction of protein structure;

(2)          It causes the breakage of DNA bases and DNA strand of microorganisms;

(3)          It promotes the transform from ADP into ATP, resulting in energy consumption.

HPP

High pressure

HPP destroys cell membranes and leads to cytoplasm loss [37], resulting in following consequences:

(1)          It induces enzyme denaturation, resulting in the destruction of enzyme structure [38];

(2)          It cause the the destruction of ribosome and promotes the transform from ADP into ATP, resulting in energy consumption [39];

(3)          The nucleic acids (DNA and RNA) inside the cell are damaged and disruption of protein biosynthesis [40].

Biological bacteria reduction technology

Plant-derived natural antimicrobials

Plant antimicrobials

Volatile components

Tannates

Aromatic compounds

It enters into cell through diffusion to disrupt microbial cell walls and cell membranes, inhibiting ATP synthesis and reducing energy metabolism

Animal-derived natural antimicrobials

Amino acids

Polymer sugars

Damage to cell walls and cell membranes results in increased membrane permeability, which affects energy conversion, synthesis of biomolecules, and disrupts cell metabolism.

Microbial-derived natural antimicrobials

Microbial metabolites

It alters the permeability of cell membranes or inhibits the growth of microorganisms through competition for nutrients.

Reviewer 3 Report

Comments and Suggestions for Authors

Comments chronologically:

l. 6 - Expected space before Zigong.

l. 7 - Expected space before Bazhong.

l. 28 - The sentence is incomprehensible. Please verify the meaning. Verify the correctness of the literature reference.

l. 36 - Unexpected space after the openinig parenthesis.

l. 40 - Expected "etc" correction.

l.40 and next - Expected space before parenthesis.

l. 50 - The theory of "fence technology" is not known. Explanation and reference to literature on "fence technology" is expected. (hurdle technology?)

l. 54-58 - The purpose of the manuscript should be explained. What is the originality of the manuscript? How does it differ from other similar ones?

The manuscript is a review. What databases were used for the review? What publication selection method was used? What criteria were used? The method must be described in detail.

l. 59 and next - The use of the words "technique" and "technology" should be corrected. They are not synonyms.

l. 73 - In chemistry, hypochlorite is an anion.

l. 86 - The statement "reasonable range" needs clarification. What documents determine "range"?

l. 89 - Not only, bocause also UV radiationand others. Clarification needed.

l. 92 - Necessary correct description of radicals.

l. 100-101 - I disagree with such a generalization. Please verify this sentence.

l. 102 - I disagree with such a generalization. For example, X-rays, HPP cause changes in taste, odour, look. It often induces free radicals, which has its consequences. Please verify the sentence.

l. 105-110 - Very lapidary. Content does not provide key information. Please expand on the description. Especially since ultrasound can also stimulate microbial growth. They have a very selective effect. Please complete, among other things, the basic parameters of the food processing process.

l. 114 - Not all attributs of "original quality". Please verify your sentence. Please complete, among other things, the basic parameters of the food processing process.

l. 129 and next - Verify the capital letter of "plasma". Other comments similar to those above.

Figure 1 should be explained and commented on more extensively.

Table 1 - Comment as above. Please clearly separate (e.g. horizontal lines) the techniques described in the table.

The "mode of action" does not mention the electron rays described in the next column. It should be clarified.

Chapter 2 on method classification is generally unnecessary. I propose to combine and supplement the contents of Chapters 2 and 3 into one describing the physical, chemical and biological basis of the methods. 

l. 188 - The authors name should be verified.

l. 184 and others, Table 1 - Please verify the correct use of capital letters, punctuation throughout the text.

l. 193 - The "duration of treatment" is only one paramter of bactericidal effect. What is the effect of concentration?

l. 193 - Please verify the reference.

l. 229 - The title should be expanded.

l. 232,251,255 and others - Italic expected in names of bacteria.

l. 233 - Verify capitalic letters.

l. 242 - Verify reference.

l. 254-255 - Parameter information should be completed. Frequency and time are completely insufficient to get the desired effect. What about the intensity?

l. 256 - Does the sterilization effect occur throughout the product? Please clarify this.

l. 260 - It is possible to reduce the number of microorganisms on the surface of meat without exceeding the threshold of ultrasonic cavitation. Please complete the content based on relevant publications.

l. 286-287 - Please verify this statement in terms of current standards (e.g. CAC) and food laws (e.g. EU).

l. 292 - Verify the author's name in the reference.

l. 293,294 - Verify the unit of measure.

l. 308 - "Fence technology" is unknown. Should be explained.

l. 318-319 - Verify the parameters.

l. 344 - The Latin name of the plant should be written in italic.

The content of Chapter 4.3 needs to supplemented. The "plant-derived", "animal-derived", and "microbial derived" antimicrobials applications description is expected according to Table 1. And what about bacteriocins?

Verify the text of Chapter 4.4. Is it about "hurdle technology"?

Chapter 4 contains only general information. It is necessary to supplement the content of the chapter, for example, with parameters for conducting an effective process. It should be completed with HPP technology, too.

Conclusions should be rewritted bocause the title and aim of manuscript.

Mismatches between references and bibliography should be removed.

The literature list contains many errors (journal names abbreviated and full names - e.g., 1 and 4, no authors - e.g., 22, no data - e.g., 51). It should be carefully reviewed for compliance with editorial requirements.

Author Response

Response to Reviewer #3

Thank you very much for the recognition of this study. We would like to express our most sincere appreciation for your careful review of this manuscript. The valuable suggestions you had proposed were beneficial to improve the quality of this manuscript. The modifications have been performed carefully according to your suggestions as follow:

Q1: l. 6 - Expected space before Zigong.

A1: Thanks for your suggestion. I’m so sorry for giving you so much trouble for the spelling errors. In the revised manuscript, the sentence has been corrected in line 6 as follow:

BaHang Food Development Co., Zigong 643000, China

Q2: l. 7 - Expected space before Bazhong.

A2: Thanks for your suggestion. I’m so sorry for giving you so much trouble for the spelling errors. In the revised manuscript, the sentence has been corrected in line 7 as follow:

SiChuan AiChiTu Food Co., Bazhong 636600, China

Q3: l. 28 - The sentence is incomprehensible. Please verify the meaning. Verify the correctness of the literature reference.

A3: Thank you for your suggestion. I 'm sorry for the grammatical errors in the sentence. In the revised manuscript, the sentence has been corrected in line 28 and the appropriate literature reference has been replaced in line 25-29 as follow:

Meat contains a large amount of fat, protein, vitamins, minerals and other nutrients[1]. These nutrients can be utilized by foodborne microbial such as Pseudomonas, Acinetobacter, Enterobacter, Lactobacillus and other microorganisms to grow and reproduce, which not only cause the decrease of raw meat character but also shorten the shelf-life of the meat [2].

[1]. Wang, J.; Chen, J.; Sun, Y.; He, J.; Zhou, C.; Xia, Q.; Dang, Y.; Pan, D.; Du, L. Ultraviolet-Radiation Technology for Preservation of Meat and Meat Products: Recent Advances and Future Trends. Food Control 2023, 148, 109684, doi:10.1016/j.foodcont.2023.109684.

[2]. Woraprayote, W.; Malila, Y.; Sorapukdee, S.; Swetwiwathana, A.; Benjakul, S.; Visessanguan, W. Bacteriocins from Lactic Acid Bacteria and Their Applications in Meat and Meat Products. Meat Science 2016, 120, 118–132, doi:10.1016/j.meatsci.2016.04.004.

Q4: l. 36 - Unexpected space after the openinig parenthesis.

A4: Thanks for your reminding. In the revised manuscript, the sentence has been corrected in line 34-37 as follow:

At present, the common bactericidal technologies at home and abroad mainly includes chemical bacteria reduction technology (slightly acidic electrolyzed water, organic acids, ozone), non-thermal physical bacteria reduction technology (ultrasound, irradiation, ultraviolet, cold plasma,

Q5: l. 40 - Expected "etc" correction.

A4: Thanks for your reminding. In the revised manuscript, the sentence has been corrected in line 38-39 as follow:

(plant-derived natural antibacterial agents, animal-derived natural antibacterial agents, microbial source natural antibacterial agents, etc. [3]).

Q6: l.40 and next - Expected space before parenthesis.

A4: Thanks for your reminding. In the revised manuscript, the sentence has been corrected in line 38-39 as follow:

(plant-derived natural antibacterial agents, animal-derived natural antibacterial agents, microbial source natural antibacterial agents, etc. [3]).

Q7: l. 50 - The theory of "fence technology" is not known. Explanation and reference to literature on "fence technology" is expected. (hurdle technology?)

A7: Thanks for your suggestion. I am so sorry that the theory of "fence technology" was not clearly described. The hurdle technology is a method for ensuring the safety of food products by controlling or destroying microbial spoilage or food pathogens through the combination of a series of preservative parameters (hurdles) imposing physiological impact on microbial cells (Giannakourou, Leistner et al., 2021). Hurdle technology uses a combination of preservation methods, such as controlling the temperature, decreasing water activity, and modification of acidity, to ensure food safety and extend the shelf life of products with minimal processing. Many researchers utilize physical sterilization technology, chemical bactericidal technology, biological bactericidal technology, and low temperature preservation technology as a combination of defense factors to achieve the preservation effect of '1+1 > 2' through fence technology. Mikš -Krajnik et al. (Mikš -Krajnik et al., 2016) pointed out that SAEW alone was not sufficient for complete inactivation of microorganisms. However, combining SAEW with ultrasound (Li et al., 2017) has been shown to provide rapid and effective bactericidal results. Similarly, ozone has been combined with fencing factors such as irradiation (Taiye Mustapha et al., 2020), organic acids (Pounraj et al., 2021), and ultraviolet light to control microbial growth and extend the shelf life of food. In the process of meat storage, it is necessary to scientifically select the appropriate bacteria-reducing technology based on the type of meat, processing methods, and storage conditions. In the revised manuscript, the sentence has been corrected in line 46-54 as follow:

However, the application of single bacterial reduction technology may reduce the antibacterial effect due to the induced stress response of bacteria, while the combined application of multiple bacteria-reducing technologies can avoid the above drawbacks to a certain extent. The hurdle technology is a method for ensuring the safety of food products by controlling or destroying microbial spoilage or food pathogens through the combination of a series of preservative parameters (hurdles) imposing physiological impact on microbial cells [4,5]. Many researchers utilize physical sterilization technology, chemical bactericidal technology, biological bactericidal technology, and low temperature preservation technology as a combination of defense factors to achieve the preservation effect of '1+1 > 2' through fence technology [6–9]. In the process of meat storage, it is necessary to scientifically select the appropriate bacteria-reducing technology based on the type of meat, processing methods, and storage conditions.

[4]. Giannakourou, M.C.; Tsironi, T.N. Application of Processing and Packaging Hurdles for Fresh-Cut Fruits and Vegetables Preservation. Foods 2021, 10, 830, doi:10.3390/foods10040830.

[5]. Leistner, L. Basic Aspects of Food Preservation by Hurdle Technology. International Journal of Food Microbiology 2000, 55, 181-186, doi:10.1016/S0168-1605(00)00161-6.

[6]. Mikš-Krajnik, M.; James Feng, L.X.; Bang, W.S.; Yuk, H.-G. Inactivation of Listeria Monocytogenes and Natural Microbiota on Raw Salmon Fillets Using Acidic Electrolyzed Water, Ultraviolet Light or/and Ultrasounds. Food Control 2017, 74, 54–60, 535 doi:10.1016/j.foodcont.2016.11.033. 536

[7]. Li, J.; Ding, T.; Liao, X.; Chen, S.; Ye, X.; Liu, D. Synergetic Effects of Ultrasound and Slightly Acidic Electrolyzed Water against Staphylococcus Aureus Evaluated by Flow Cytometry and Electron Microscopy. Ultrason Sonochem 2017, 38, 711–719.doi:10.1016/j.ultsonch.2016.08.029.

[8]. Pounraj, S.; Bhilwadikar, T.; Manivannan, S.; Rastogi, N.K.; Negi, P.S. Effect of Ozone, Lactic Acid and Combination Treatments on the Control of Microbial and Pesticide Contaminants of Fresh Vegetables. Journal of the Science of Food and Agriculture 2021, 101, 3422–3428, doi:10.1002/jsfa.10972. 545.

[9]. Taiye Mustapha, A.; Zhou, C.; Wahia, H.; Amanor-Atiemoh, R.; Otu, P.; Qudus, A.; Abiola Fakayode, O.; Ma, H. Sonozonation: Enhancing the Antimicrobial Efficiency of Aqueous Ozone Washing Techniques on Cherry Tomato. Ultrason Sonochem 2020, 64, 105059, doi:10.1016/j.ultsonch.2020.105059.

Giannakourou, M.C.; Tsironi, T.N. Application of Processing and Packaging Hurdles for Fresh-Cut Fruits and Vegetables Preservation. Foods 2021, 10, 830, doi:10.3390/foods10040830.

Leistner, L. Basic Aspects of Food Preservation by Hurdle Technology. International Journal of Food Microbiology 2000, 55, 181-186, doi:10.1016/S0168-1605(00)00161-6.

Mikš-Krajnik, M.; James Feng, L.X.; Bang, W.S.; Yuk, H.-G. Inactivation of Listeria Monocytogenes and Natural Microbiota on Raw Salmon Fillets Using Acidic Electrolyzed Water, Ultraviolet Light or/and Ultrasounds. Food Control 2017, 74, 54–60, 535 doi:10.1016/j.foodcont.2016.11.033. 536

Li, J.; Ding, T.; Liao, X.; Chen, S.; Ye, X.; Liu, D. Synergetic Effects of Ultrasound and Slightly Acidic Electrolyzed Water against Staphylococcus Aureus Evaluated by Flow Cytometry and Electron Microscopy. Ultrason Sonochem 2017, 38, 711–719.doi:10.1016/j.ultsonch.2016.08.029.

Taiye Mustapha, A.; Zhou, C.; Wahia, H.; Amanor-Atiemoh, R.; Otu, P.; Qudus, A.; Abiola Fakayode, O.; Ma, H. Sonozonation: Enhancing the Antimicrobial Efficiency of Aqueous Ozone Washing Techniques on Cherry Tomato. Ultrason Sonochem 2020, 64, 105059, doi:10.1016/j.ultsonch.2020.105059.

Pounraj, S.; Bhilwadikar, T.; Manivannan, S.; Rastogi, N.K.; Negi, P.S. Effect of Ozone, Lactic Acid and Combination Treatments on the Control of Microbial and Pesticide Contaminants of Fresh Vegetables. Journal of the Science of Food and Agriculture 2021, 101, 3422–3428, doi:10.1002/jsfa.10972. 545

Perez, S.L.; Chianfrone, D.J.; Bagnato, V.S.; Blanco, K.C. Optical Technologies for Antibacterial Control of Fresh Meat on Display. LWT 2022, 160, 113213, doi:10.1016/j.lwt.2022.113213.

Q8: l. 54-58 - The purpose of the manuscript should be explained. What is the originality of the manuscript? How does it differ from other similar ones?

A8: Thank you very much for your reminding. In this paper, several commonly bacteria-reducing pretreatment technologies were reviewed, the mechanisms of bacterial inhibition were analyzed, and the application progress in meat preservation was summarized. Furthermore, the advantages and disadvantages of these bacteria-reducing technologies as well as their future development trend were discussed. This paper aimed to provide a scientific theoretical reference for better understanding and application of bacteria-reducing technologies in meat preservation.

In terms of originality and differences from other similar ones, the approximately eleven papers related with bacteria-reducing pretreatment technology were indexed in recent years. The title of these papers were listed:

  • Ultraviolet-radiation technology for preservation of meat and meat products: Recent advances and future trends
  • Application of ultrasound technology in processing of ready-to-eat fresh food: A review
  • Ultrasound technology as inactivation method for foodborne pathogens: A Review
  • The use of ozone technology to control microorganism growth, enhance food safety and extend shelf life: A promising food decontamination technology
  • Ozone based food preservation: a promising green technology for enhanced food safety
  • Inactivation of foodborne viruses by high-pressure processing (HPP)
  • Microbiological aspects of high-pressure processing of food: Inactivation of microbial vegetative cells and spores
  • A review on novel non‐thermal food processing techniques for mycotoxin reduction
  • Intervention technologies for ensuring microbiological safety of meat: Current and future trends
  • A comprehensive review on impact of non-thermal processing on the structural changes of food components
  • A review on recent development in non-conventional food sterilization technologies

Moat of papers have focused on the application of single bacteriological reduction technology in food. Only four papers introduced physical bacteria-reducing technologies. However, the manuscript discussed the bacteria-reducing technologies including bacteria-reducing technologies (SAEW, organic acids, ozone), non-thermal physical technologies (ultrasound, irradiation, ultraviolet, cold plasma, HPP), biological technology, and hurdle technology. Meanwhile, the mechanisms of inhibiting bacterial growth were also introducted. The differences between these papers and our work were summarized as follow:

1

2

3

4

5

6

Topic

Ultraviolet-radiation technology for preservation of meat and meat products: Recent advances and future trends

Application of ultrasound technology in processing of ready-to-eat fresh food: A review

Ultrasound technology as inactivation method for foodborne pathogens: A Review

The use of ozone technology to control microorganism growth, enhance food safety and extend shelf life: A promising food decontamination technology

Ozone based food preservation: a promising green technology for enhanced food safety

Inactivation of foodborne viruses by high-pressure processing (HPP)

Journal

(Time)

Food Control(2023)

Ultrasonics Sonochemistry(2020)

Foods(2023)

Foods(2023)

Ozone: Science & Engineering(2019)

Foods(2021)

Technology

Ultraviolet-radiation

Ultrasound

Ozone

HPP

7

8

9

10

11

Our research

Topic

Microbiological aspects of high-pressure processing of food: Inactivation of microbial vegetative cells and spores

A review on novel non‐thermal food processing techniques for mycotoxin reduction

Intervention technologies for ensuring microbiological safety of meat: Current and future trends

A comprehensive review on impact of non-thermal processing on the structural changes of food components

A review on recent development in non-conventional food sterilization technologies

Research progress on bacteria-reducing pretreatment technology of meat

Journal

(Time)

High Pressure Processing

of Food(2016)

International Journal of Food Science & Technology(2020)

Comprehensive Reviews in Food Science and Food Safety(2014)

Food Research International(2021)

Journal of Food Engineering(2016)

Foods

Technology

HPP

High pressure;

Pulsed electric filed;

Cold plasma;

Ultrasound

High hydrostatic pressure (HHP);

Irradiation;

Natural antimicrobials;

Microwave;

Radio frequency;

Pulsed electric field treatment

Cold plasma treatment;

Irradiation;

High-pressure;

Ultrasonication;

Ulsed light;

High voltage electric field;

Pulsed electric field

High pressure;

UV;

Pulsed light;

Ultrasonic;

Pulsed electric field;

Irradiation;

Ultrasound

Chemical technology

1.1 SAEW;

1.2 Organic acids;

1.3 Ozone;

Non-thermal technology

Ultrasound;

Irradiation;

Ultraviolet;

Cold plasma;

HPP;

Biological technology

Hurdle technology

In the revised manuscript, the originality, the differ from other similar paper, and the purpose of this manuscript have been added in line 55-64 as follow:

Previous studies have only focused on the application of single bacteriological reduction technology in food [10–12]. However, there has been a lack of research on the application of different bacteriological reduction technologies in meat preservation in recent years. This paper reviews several commonly bacteria-reducing pretreatment technology, analyzes the mechanisms of bacterial inhibition, and summarizes their application in meat preservation. In this paper, several commonly bacteria-reducing pretreatment technologies were reviewed, the mechanisms of bacterial inhibition were analyzed, and the application progress in meat preservation was summarized. Furthermore, the advantages and disadvantages of these bacteria-reducing technologies as well as their future development trend were discessued. This paper aimed to provide a scientific theoretical reference for better understanding and application of bacteria-reducing technologies in meat preservation.

[10]. Govaris, A.; Pexara, A. Inactivation of Foodborne Viruses by High-Pressure Processing (HPP). Foods 2021, 10, 215, doi:10.3390/foods10020215.

[11]. Chen, F.; Zhang, M.; Yang, C. Application of Ultrasound Technology in Processing of Ready-to-Eat Fresh Food: A Review. Ultrason Sonochem 2020, 63, 104953, doi:10.1016/j.ultsonch.2019.104953.

[12]. Pandiselvam, R.; Subhashini, S.; Banuu Priya, E.P.; Kothakota, A.; Ramesh, S.V.; Shahir, S. Ozone Based Food Preservation: A Promising Green Technology for Enhanced Food Safety. Ozone: Science & Engineering 2019, 41, 17–34, doi:10.1080/01919512.2018.1490636.

Q9: The manuscript is a review. What databases were used for the review? What publication selection method was used? What criteria were used? The method must be described in detail.

A9: Thanks for your suggestion. The databases, publication selection methods and screening criteria used in this review are described in detail below:

This manuscript was systematically searched for articles using databases such as Web of Science, Scopus, PubMed, and Science Direct. The publication selection methods inclued keyword searching method, journal searching method, and citation literature searching method. This paper focused on the analysis of the individual and combined effects of common meat reduction techniques on microbial inactivation. The main keywords used for the search included "bacteria reduction technology," "chemical bacteria reduction technology," "non-thermal physical bacteria reduction technology," "slightly acidic electrolytic water," "organic acid," "ozone," "ultrasonic," "irradiation," "ultraviolet," "low temperature plasma," "high pressure treatment," "natural bacteria reduction agent," "biological bacteria reduction agent ," plant bacteria reduction agent", animal bacteria reduction agent", microbial source reduction bactericide", and"combined sterilization". The journal searched included foods, food chemistry, meat science, international journal of food microbiology, and others. In additon, the title or author of the target article in the search column of SCI database was entered, and the paper was searched according to the situation that this article has been cited by other articles, so as to understand the research hot spots and frontiers in this field.

Q10: l. 59 and next - The use of the words "technique" and "technology" should be corrected. They are not synonyms.

A10: Thanks for your reminding. In the revised manuscript, these errors have been corrected carefully as follow:

(1) The sentence in line 24 has been corrected in line 39-41 line as follow:

These technologies for reducing bacteria can temporarily or permanently disrupt the living con ditions of microorganisms by altering the internal state or external environment of meat.

(2) The sentence in line 24 has been corrected in line 65-66 line as follow:

Classification of bacteria-reducing technologies and their mechanisms for reducing bacteria.

(3) The sentence in line 24 has been corrected in line 201-203 line as follow:

Natural antimicrobial agents refer to a class of bactericidal substances with complex structures that are extracted from plants and animals in nature through bioengineering technologies, or produced through microbial metabolism [31].

(4) The sentence in line 24 has been corrected in line 216-218 line as follow:

Animal-derived bacteriostatic agents refer to substances with bacteriostatic activity extracted or synthesized from animals, which are generally obtained from various tissues, secretions, or bioactive substances synthesized through bioengineering technologies.

Q11: l. 73 - In chemistry, hypochlorite is an anion.

A11: Thanks for your reminding. I am so sorry that the word of HClO was written wrong. In fact, it has a strong bactericidal effect and can make the microbial mortality rate of more than 90 % are HClO [5]. In the revised manuscript, the sentence has been corrected in line 80-81 as follow:

It has been found that the HClO in SAEW, which exhibits a highly potent bactericidal effect, can result in a microbial lethality rate of over 90 % [14].

[14]. Sun, J.; Jiang, X.; Chen, Y.; Lin, M.; Tang, J.; Lin, Q.; Fang, L.; Li, M.; Hung, Y.-C.; Lin, H. Recent Trends and Applications of Electrolyzed Oxidizing Water in Fresh Foodstuff Preservation and Safety Control. Food Chemistry 2022, 369, 130873, doi:10.1016/j.foodchem.2021.130873.

Q12: l. 86 - The statement "reasonable range" needs clarification. What documents determine "range"?

A12: Thanks for your reminding. In the revised manuscript, the sentence has been corrected in line 93-101 as follow:

Organic acids commonly used for reducing bacteria in food include lactic acid, propionic acid, sorbic acid, acetic acid, citric acid, and tartaric acid. These acids are primarily produced through microbial fermentation engineering or naturally occurring processes. They are widely employed in food preservation due to their low cost, natural coloration, high bactericidal efficiency, and international recognition as safe (GRAS) bactericidal agents [15]. In addition to their role as bacterial-reducing agents in food, organic acids are also listed in FDA regulations for various technical applications such as acidity regulators, antioxidants, flavor enhancers, and pH adjusters [16].

[15]. Smulders, F.J.M.; Greer, G.G. Integrating Microbial Decontamination with Organic Acids in HACCP Programmes for Muscle Foods: Prospects and Controversies. International Journal of Food Microbiology 1998, 44, 149–169, doi:10.1016/S0168-1605(98)00123-8.

[16]. Pohlman, F.; Dias-Morse, P.; Pinidiya, D. Product Safety and Color Characteristics of Ground Beef Processed from Beef Trimmings Treated with Peroxyacetic Acid Alone or Followed by Novel Organic Acids. Journal of microbiology, biotechnology and food sciences 2014, 4, 93–101, doi:10.15414/jmbfs.2014.4.2.93-101.

Q13: l. 89 - Not only, bocause also UV radiationand others. Clarification needed.

A13: Thanks for your suggestion. In the revised manuscript, the sentence has been corrected in line 104-106 as follow:

Ozone (O3) is a light blue gas with a distinct odor, which is produced by an ozone generator through high voltage discharge or ultraviolet radiation alone or in combination with nitrogen oxides (NOx) [17]

[17]. Kaur, K.; Pandiselvam, R.; Kothakota, A.; Padma Ishwarya, S.; Zalpouri, R.; Mahanti, N.K. Impact of Ozone Treatment on Food Polyphenols – A Comprehensive Review. Food Control 2022, 142, 109207, doi:10.1016/j.foodcont.2022.109207.

Q14: l. 92 - Necessary correct description of radicals.

A14: Thanks for your reminding. I am so sorry that the writing of radicals is wrong. In the revised manuscript, the sentence has been corrected in line 109-111 as follow:

It is also easy to decompose into HO- and O2 in water, which has a strong disinfecting and sterilising ability [19]

[19]. Xue, W.; Macleod, J.; Blaxland, J. The Use of Ozone Technology to Control Microorganism Growth, Enhance Food Safety and Extend Shelf Life: A Promising Food Decontamination Technology. Foods 2023, 12, 814, doi:10.3390/foods12040814.

Q15: l. 100-101 - I disagree with such a generalization. Please verify this sentence.

A15: Thanks for your reminding. We are very sorry for your misunderstanding caused by our lack of clear description., in the revised manuscript, the sentence has been corrected in line 122-124 as follow:

Compared with the traditional thermal sterilization technology, usually at low temperature or room temperature can achieve the purpose of sterilization, can better ensure the quality of food.

Q16: l. 102 - I disagree with such a generalization. For example, X-rays, HPP cause changes in taste, odour, look. It often induces free radicals, which has its consequences. Please verify the sentence.

A16: Thanks for your suggestion. We are very sorry for your misunderstanding caused by our lack of clear description., in the revised manuscript, the sentence has been corrected in line 122-124 as follow:

Compared with the traditional thermal sterilization technology, usually at low temperature or room temperature can achieve the purpose of sterilization, can better ensure the quality of food.

Q17: l. 105-110 - Very lapidary. Content does not provide key information. Please expand on the description. Especially since ultrasound can also stimulate microbial growth. They have a very selective effect. Please complete, among other things, the basic parameters of the food processing process.

A17: Thanks for your suggestion. In the revised manuscript, the sentence has been corrected in line 126-139 as follow:

Ultrasound is a longitudinal mechanical wave beyond the range of human hearing, with a frequency generally between 20 kHz and 1 GHz [21]. It can travel through air, liquids, and solids and has the capability to kill certain microorganisms in food. In the food industry, ultrasound at a lower frequency (20 ~ 100 KHz) is usually used to deactivate microorganisms. The bactericidal mechanism of ultrasonic waves is mainly attributed to the cavitation effect [22]. Under the influence of ultrasonic waves, cavitation bubbles oscillate, grow, and burst, resulting in instantaneous high temperature and pressure. The rapid alternation of temperature and pressure directly damages the cell wall or membrane of microorganisms, promotes water molecule decomposition, and triggers free radical reactions. These generated free radicals possess strong oxidizing properties that damage the DNA and enzymes of microbial cells, leading to microbial deactivation [23]. The technology possesses characteristics such as low cost, green safety, wide applicability, and convenience. However, the penetration ability of ultrasonic waves is limited; therefore, it is commonly combined with other bacterial-reducing technologies in practical applications.

[22]. Lin, L.; Wang, X.; Li, C.; Cui, H. Inactivation Mechanism of E. Coli O157:H7 under Ultrasonic Sterilization. Ultrasonics Sonochemistry 2019, 59, 104751, doi:10.1016/j.ultsonch.2019.104751.

[23]. Liao, X.; Li, J.; Suo, Y.; Chen, S.; Ye, X.; Liu, D.; Ding, T. Multiple Action Sites of Ultrasound on Escherichia Coli and Staphylococcus Aureus. Food Science and Human Wellness 2018, 7, 102–109, doi:10.1016/j.fshw.2018.01.002.

Q18: l. 114 - Not all attributs of "original quality". Please verify your sentence. Please complete, among other things, the basic parameters of the food processing process.

A18: Thanks for your suggestion. In the revised manuscript, the sentence has been corrected in line 140-151 as follow:

Irradiation sterilisation technology utilizes ionising radiation sources of various wavelengths, such as electron beams, γ-rays or X-rays to radiate food [24], reducing or removing most of the harmful microorganisms in food. This technology can minimize the decline in food quality, extend shelf life, and is widely used in the field of food processing and preservation. After irradiation, microorganisms in food (such as bacteria, yeast, and mold) absorb radiation energy which breaks chemical bonds and changes cytochemical composition to achieve sterilization [24]. In general, to ensure the safety of processed meat, the maximum absorbed dose of irradiated food should not exceed 10 kGy according to the standards established by the Codex Alimentarius Commission (CAC) [25]. However, different countries have varying standards for using irradiation in food. In the food industry, it is also necessary to determine the irradiation dose based on different types of food in order to effectively kill harmful microorganisms while maintaining food quality.

[24]. Zhang, J.; Zhang, Q.; Li, F. Characteristics of Key Microorganisms and Metabolites in Irradiated Marbled Beef. Meat Science 2023, 199, 109121, doi:10.1016/j.meatsci.2023.109121.

[25]. General Standard for Irradiated Foods. Codex Stan 2003, 106-1983, Rev.1 - 2003.

Q19: l. 129 and next - Verify the capital letter of "plasma". Other comments similar to those above.

A19: Thanks for your reminding. In the revised manuscript, the sentence has been corrected as follow:

  • The sentence has been corrected in line 174-176 as follow:

Cold plasma is a type of electroneutral ionized gas, which is composed of electrons, positive and negative ions, reactive oxygen species, excited or non-excited gas molecules and photons.

(2)  The sentence has been corrected in line 445-450 line as follow:

3.2.4. Cold plasma

In recent years, Cold plasma has gained widespread attention in the field of food sterilization and shows great potential in meat preservation applications. It can effectively eliminate various types of viruses, bacteria, fungi, and spores. The ability of Cold plasma to inhibit bacteria mainly depends on factors such as gas composition, airflow, electrical input, and duration of the process[72].

Q20: Figure 1 should be explained and commented on more extensively.

A20: Thanks for your suggestion. In the revised manuscript, the Figure 1 has been redrawn, the mechanism of bacteria-reducing technology of meat has been also supplemented in Table in line 251 as follow:

Table 2. Mechanism of bacteriostasis in meat reducing technology

Bacteriological reduction technology

Categorisation

Mode of action

Antibacterial mechanism

Chemical bacteria reduction technology

SAEW

HCIO

CIO-

Reactive oxygen species

The cell membrane is damaged by SAEW, causing rapid leakage of K+ and an increase in membrane permeability. This causes HClO and ClO- to enter the cell, resulting in the following consequences:

(1)   It causes the breakage of DNA base, which inhibits the protein biosynthesis;

(2)   It causes the breakage of protein structure;

(3)   It causes changes in electron flow, inhibiting microbial energy metabolism and ATP production. [32]

Organic acids

RCOOH

COOH-

RCOOH enters into the cell, leading to the following consequences:

(1)  RCOOH· ionized by RCOOH causes the breakage of DNA base and DNA strand, which inhibits the protein biosynthesis;

(2)  The H+ ionized by RCOOH reduce the pH value of the cells, and H+is excreted out of the cell through energy released by the transformation from ATP into ADP. The consumption of a large amount of energy inhibits the reproduction of bacteria.

Ozone

O3

O3 increases the permeability of the cell membrane and destroys lipoproteins and lipopolysaccharides, resulting in the following results after entering the cell:

(1)  It causes the breakage of enzyme structure;

(2)  It disrupts the structures of DNA and RNA, interfering with protein synthesis. [19,33]

Non-thermal physical bacteria reduction technology

Ultrasound

Cavitation effect

Reactive oxygen species

Microbial cells experience violent oscillations that disrupt the permeability of cell membranes and release reactive oxygen species enter the cell, which results following consequences:

(1)  The water molecules in the liquid medium decompose into reactive oxygen species, causing reduction reactions and secondary oxidation, which results in changes to the structure of the solute; [34]

(2)   It causes the breakage of enzyme structure. [35]

Radiation

Electron rays

γ-rays

X-rays

(1)  Electron rays damage the inner cell membrane, leading to disruption of the enzyme system and impairment of functions;

(2)  It causes the breakage of DNA bases of microorganisms. [36]

UV

UV

It destroys the DNA base, inhibits DNA transcription, replication and cell division, and inhibits protein synthesis by altering or destroying the structure of DNA or RNA molecules. [1]

Cold plasma

Active substances

Reactive oxygen species

Electrically charged particles

Reactive Oxygen Species (ROS), Reactive Nitrogen Species (RNS), and charged particles destroy bacterial cells and then enter the cell interior, resulting in the following consequences:

(1)  It causes the destruction of protein structure;

(2)  It causes the breakage of DNA bases and DNA strand of microorganisms;

(3)  It promotes the transform from ADP into ATP, resulting in energy consumption.

HPP

High pressure

HPP destroys cell membranes and leads to cytoplasm loss [37], resulting in following consequences:

(1)  It induces enzyme denaturation, resulting in the destruction of enzyme structure; [38]

(2)  It cause the the destruction of ribosome and promotes the transform from ADP into ATP, resulting in energy consumption; [39]

(3)  The nucleic acids (DNA and RNA) inside the cell are damaged and disruption of protein biosynthesis. [40]

Biological bacteria reduction technology

Plant-derived natural antimicrobials

Plant antimicrobials

Volatile components

Tannates

Aromatic compounds

It enters into cell through diffusion to disrupt microbial cell walls and cell membranes, inhibiting ATP synthesis and reducing energy metabolism

Animal-derived natural antimicrobials

Amino acids

Polymer sugars

Damage to cell walls and cell membranes results in increased membrane permeability, which affects energy conversion, synthesis of biomolecules, and disrupts cell metabolism.

Microbial-derived natural antimicrobials

Microbial metabolites

It alters the permeability of cell membranes or inhibits the growth of microorganisms through competition for nutrients.

Q21: Table 1 - Comment as above. Please clearly separate (e.g. horizontal lines) the techniques described in the table.

A21: Thanks for your suggestion. In the revised manuscript, the horizontal lines used to clearly separate the techniques has been added in Table 1 in line 251 just as A20.

Q22: The "mode of action" does not mention the electron rays described in the next column. It should be clarified.

A22: Thanks for your suggestion. The revised manuscript has been supplemented as shown in Table 1

Q23: Chapter 2 on method classification is generally unnecessary. I propose to combine and supplement the contents of Chapters 2 and 3 into one describing the physical, chemical and biological basis of the methods.

A23: Thanks for your suggestion. In the revised manuscript, the contents of the original Chapters 2 and 3 have been merged into the current Chapter 2 (line 65-253) as follow:

  1. Classification of bacteria-reducing technologies and their mechanisms for reducing bacteria

2.1 Chemical bacteria reduction technology

2.2 Non-thermal physical bacteria reduction technology

2.3 Biological bacteria reduction technology

2.4. Mechanisms of bacterial inhibition in meat reduction techniques

  1. Application of bacteriological reduction technologies in meat

3.1. Application of chemical bacteria reduction technology in meat

3.1.1. SAEW

3.1.2. Organic acids

3.1.3. Ozone

3.2. Application of non-thermal physical bacteria reduction technology in meat

3.2.1. Ultrasound

3.2.2. Irradiation

3.2.3. Ultraviolet

3.2.4. Cold plasma

3.2.5. HPP

3.3. Application of biological bacteria reduction technology in meat

  1. Conclusions

Q24: l. 188 - The authors name should be verified.

A24: Thanks for your reminding. In the revised manuscript, the authors name has been corrected in line 258-260 as follow:

Among these methods, soaking treatment yields the best bactericidal effect. Rahman et al. [41] soaked fresh chicken breast with SAEW...

Q25: l. 184 and others, Table 1 - Please verify the correct use of capital letters, punctuation throughout the text.

A25: Thanks for your reminding. In the revised manuscript, the incorrect use of capital letters in line 184, others, and Table 1 of the initial manusript has been rectified as follow:

(1) The incorrect use of capital letters in line 184 has been rectified in line 255 as follow:

3.1. Application of chemical bacteria reduction technology in Meat

(2) The other incorrect use of capital letters was rectified in line 512 as follow:

3.3. Application of biological bacteria reduction technology in meat

(3) The incorrect use of capital letters in Table 1 have been rectified in Table 2.

Q26: l. 193 - The "duration of treatment" is only one paramter of bactericidal effect. What is the effect of concentration? Please verify the reference.

A26: Thanks for your reminding. You are right. In the revised manuscript, the sentence and the reference have been corrected in line 263-269 as follow:

Furthermore, the bactericidal effect of SAEW varies depending on the duration and concentration of treatment. Ye et al. [43] investigated the effects of different treatment times and concentrations of SAEW on raw frozen shrimp and found that treating with SAEW for 5 minutes had a better bactericidal effect than treating with SAEW for 3 minutes. Additionally, under the same treatment time, the bactericidal effect of 29 mg/L of SAEW was significantly enhanced compared to that of 16 mg/L of SAEW.

[43]. Ye, Z.; Qi, F.; Pei, L.; Shen, Y.; Zhu, S.; He, D.; Wu, X.; Ruan, Y.; He, J. Using Slightly Acidic Electrolyzed Water for Inactivation and Preservation of Raw Frozen Shrimp (Litopenaeus Vannamei) in the Field Processing. Applied Engineering in Agriculture 2014, 935–941, doi:10.13031/aea.30.10657.

Q27: l. 229 - The title should be expanded.

A27:Thanks for your reminding. In the revised manuscript, the sentence has been corrected in line 316 as follow:

3.1.3. Ozone

Q28: l. 232,251,255 and others - Italic expected in names of bacteria.

A28: Thanks for your reminding. In the revised manuscript, the words have been corrected in line 318-320, 355-358 as follow:

(1) The sentence has been corrected in line 318-320 line as follow:

Its safety and effectiveness have been proven for killing a variety of foodborne pathogens in food, such as Escherichia coli, Salmonella, and Listeria monocytogenes [50].

  • The sentence has been corrected in line 355-358 line as follow:

Morild et al. [58] evaluated the effect of pressurized steam combined with high-power ultrasound on pathogen inactivation on pig skin and pork surface. The study examined the inactivation of Salmonella typhimurium, Salmonella Derby, Salmonella infantile, Yersinia enterocolitica, and a non-pathogenic E. coli.

Q29: l. 233 - Verify capitalic letters.

A29: Thanks for your reminding. In the revised manuscript, the spelling of author has been corrected in line 320 as follow:

Cárdenas et al. [51] analyzed the antibacterial effect of gaseous ozone on chilled fresh

Q30: l. 242 - Verify reference.

A30: Thanks for your reminding. In the revised manuscript, the spelling of author has been corrected in line 333-339 as follow:

Ayranci et al. [54] discovered that ozone treatment (1 x 10-2 kg m-3, 8 hours) had detrimental effects on the physicochemical properties and color of turkey breast. The high oxidation potential of ozone may be the primary reason for lipid and protein oxidation in meat. Additionally, it was observed that the connective tissue membrane with a high protein content in turkey meat underwent denaturation after ozone treatment, which could explain the lighter coloration of turkey meat.

Q31: l. 254-255 - Parameter information should be completed. Frequency and time are completely insufficient to get the desired effect. What about the intensity?  l. 256 - Does the sterilization effect occur throughout the product? Please clarify this.

A31: Thanks for your reminding. In the revised manuscript, the sentence has been supplemented in line 349-354 as follow:

Huu, et al. [57] used ultrasonic treatments with a frequency of 40 kHz and a power density of 0.092 W/mL for E. coli O157:H7 or L. innocua. The initial bacterial content was 1 × 106 CFU/mL, and the bacterial content remained at the initial level after both 30 min and 45 min of ultrasonic treatment. The results showed that there was no significant influence on either bacteria in the product during the entire period of ultrasonic treatment. The results showed that there was no significant influence on either bacteria in the product during the entire period of ultrasonic treatment.

[57]. Nguyen Huu, C.; Rai, R.; Yang, X.; Tikekar, R.V.; Nitin, N. Synergistic Inactivation of Bacteria Based on a Combination of Low Frequency, Low-Intensity Ultrasound and a Food Grade Antioxidant. Ultrasonics Sonochemistry 2021, 74, 105567, doi:10.1016/j.ultsonch.2021.105567.

Q32: l. 260 - It is possible to reduce the number of microorganisms on the surface of meat without exceeding the threshold of ultrasonic cavitation. Please complete the content based on relevant publications.

A32: Thanks for your reminding. In the revised manuscript, the sentence has been corrected in line 346-347 as follow:

In the food industry, high-power ultrasonic waves with a frequency of 20 ~ 100 kHz are generally used to inactivate microorganisms [55].

[55]. Chen, J.H.; Ren, Y.; Seow, J.; Liu, T.; Bang, W.S.; Yuk, H.G. Intervention Technologies for Ensuring Microbiological Safety of Meat: Current and Future Trends. Comprehensive Reviews in Food Science and Food Safety 2012, 11, 119–132, doi:10.1111/j.1541-4337.2011.00177.x.

Q33: l. 286-287 - Please verify this statement in terms of current standards (e.g. CAC) and food laws (e.g. EU).

A33: Thanks for your reminding. I am so sorry that the describe about the irradiation treatmentIn is not precise.

Irradiation treatment is considered as an approved non-thermal technique by the Food and Drug Administration (FDA), World Health Organization (WHO), International Atomic Energy Agency (IAEA), and the United States Department of Agriculture (USDA) to reduce and/or eliminate microorganisms in cereal grains to improve their safety (Lorenz and Miller, 1975; Mollakhalili Meybodi et al., 2017; Pillai and Shayanfar, 2015; Zuleta et al., 2006). Meanwhile, food irradiation has been approved by various official international organizations, e.g., FAO, IAEA, and WHO, as a safe and effective method (Ito et al., 2018); only three certain types of ionizing radiation can be used for the treatment of foods by the Codex General Standard for Irradiated Foods (CODEX STAN 106–1983, REV.1-2003). The permitted irradiation treatments approved by Codex General standard consist of gamma rays (γ-rays) from the radionuclides cobalt-60 (Co-60) or cesium-137 (Cs-137), X-rays generated from machine sources operated at or below an energy level of 5 MeV and electrons (e-beams) generated from machine sources operated at or below an energy level of 10 MeV (Hazards, 2011).

  1. Lorenz, B.S. Miller Irradiation of cereal grains and cereal grain products Crit. Rev. Food Sci. Nutr., 6 (4) (1975), pp. 317-382, 10.1080/10408397509527195.
  2. Mollakhalili Meybodi, M.A. Mohammadifar, M. Farhoodi, J.L. Skytte, K. Abdolmaleki Physical stability of oil-in-water emulsions in the presence of gamma irradiated gum tragacanth J. Dispersion Sci. Technol., 38 (6) (2017), pp. 909-916
  3. Pillai, S. Shayanfar Introduction to electron beam pasteurization in food processing. S.S.S. Pillai (Ed.), Electron Beam Pasteurization and Complementary Food Processing Technologies, Elsevier (2015), pp. 3-9, 10.1533/9781782421085.1.3
  4. Zuleta, L. Dyner, M.E. Sambucetti, A. de Francisco Effect of gamma irradiation on the functional and nutritive properties of rice flours from different cultivars Cereal Chem., 83 (1) (2006), pp. 76-79, 10.1094/CC-83-0076

V.C. Ito, C.D. Bet, J.P. Wojeicchowski, I.M. Demiate, M.H.F. Spoto, E. Schnitzler, L.G. Lacerda. Effects of gamma radiation on the thermoanalytical, structural and pasting properties of black rice (Oryza sativa L.) flour. J. Therm. Anal. Calorim., 133 (1) (2018), pp. 529-537

E.P.o.B. Hazards. Scientific Opinion on the efficacy and microbiological safety of irradiation of food. EFSA J., 9 (4) (2011), p. 2103, 10.2903/j.efsa.2011.2103

Q34: l. 292 - Verify the author's name in the reference.

A34: Thanks for your reminding. In the revised manuscript, the sentence has been corrected in line 413-415 as follow:

In the study conducted by Söbeli et al. [66], various doses of UV were used to irradiate beef fillet, and it was observed that a higher dose (4.2 J/cm2) significantly reduced the total number of aerobic bacteria by 3.49±0.67 log CFU/g.

Q35: l. 293,294 - Verify the unit of measure.

A35: Thanks for your reminding. In the revised manuscript, the sentence has been corrected in line 413-415 as follow:

In the study conducted by Söbeli et al. [66], various doses of UV were used to irradiate beef fillet, and it was observed that a higher dose (4.2 J/cm2) significantly reduced the total number of aerobic bacteria by 3.49±0.67 log CFU/g.

Q36: l. 308 - "Fence technology" is unknown. Should be explained.

A36: Thanks for your reminding. In the revised manuscript, the sentence has been corrected in line 46-52 as follow:

The hurdle technology is a method for ensuring the safety of food products by controlling or destroying microbial spoilage or food pathogens through the combination of a series of preservative parameters (hurdles) imposing physiological impact on microbial cells [4,5]. Many researchers utilize physical sterilization technology, chemical bactericidal technology, biological bactericidal technology, and low temperature preservation technology as a combination of defense factors to achieve the preservation effect of “1+1 > 2” through fence technology [6–9].

Q37: l. 318-319 - Verify the parameters.

A37: Thanks for your reminding. In the revised manuscript, the sentence has been corrected in line 450-458 as follow:

Kim et al. [73] used helium (10 lpm) and a mixture of helium and oxygen (10 lpm and 10 sccm) as the excitation medium to process sliced bacon under the conditions of input power at 75, 100, and 125 W, with treatment times of 60 and 90 seconds respectively. After plasma helium treatment and helium/oxygen mixed treatment, the total number of aerobic bacteria decreased by 1.89 decimals and 4.58 decimals respectively. The study also found that regardless of gas composition, the bacteria-reducing effect is enhanced with increasing input power and treatment time. In addition, low-temperature plasma has different effects on reducing bacteria in different microbial species.

Q38: l. 344 - The Latin name of the plant should be written in italic.

A38: Thanks for your reminding. In the revised manuscript, the sentence has been corrected in line 527-530 as follow:

This was demonstrated by Wang et al. [84], who found that nanoemulsions of litsea cubeba essential oil exhibited stronger inhibition against Listeria monocytogenes and Streptococcus maritimus compared to pure litsea cubeba essential oil.

Q39: The content of Chapter 4.3 needs to supplemented. The "plant-derived", "animal-derived", and "microbial derived" antimicrobials applications description is expected according to Table 1. And what about bacteriocins?

A39: Thanks for your reminding. In the revised manuscript, the sentence has been corrected in line 513-559 as follow:

Natural antimicrobials have long been used as food additives, primarily to extend the shelf life and maintain the quality of meat and meat products. Among natural antibacterial compounds, plant-derived antimicrobials have been extensively studied, and substances such as phenols, isoflavones, ketones, acids, terpenes (essential oils), and alkaloids commonly found in meat can limit or hinder the growth of harmful microorganisms. Different types of microorganisms in meat and meat products are inhibited by various plant extracts. Studies have shown that when thyme oil and balsam oil [82] are applied to chicken breast, balsam oil significantly restricts the growth of salmonella while thyme oil effectively inhibits the growth of E. coli. Furthermore, they are incorporated into meat in various forms, such as direct mixing, coating, layering, nanoencapsulation, and microencapsulation. This can enhance its antibacterial properties and significantly expand the application of natural antimicrobials in meat. For instance, Noshad et al. [83] blended lemon essential oil with psyllium seed slime to create a novel edible coating for beef storage and preservation, and discovered that it could extend the shelf life by 10 days. Moreover, different addition technologies may have varying inhibitory effects on microorganisms. This was demonstrated by Wang et al. [84], who found that nanoemulsions of litsea cubeba essential oil exhibited stronger inhibition against Listeria monocytogenes and Streptococcus maritimus compared to pure litsea cubeba essential oil.

Chitosan, a polycationic biopolymer commonly extracted from the exoskeletons of crustaceans such as crabs and lobsters, is frequently used as an animal-derived antimicrobial agent in meat. Darmadji et al. [85] discovered that chitosan at concentrations ranging from 0.1% to 1% effectively inhibits the growth of spoilage bacteria and pathogenic bacteria during the storage of refrigerated meat. In addition to chitosan, lysozyme extracted from eggs and milk is also commonly employed. Furthermore, lactoferrin [86], a natural antibacterial substance found in secretions like saliva, milk, and tears of mammals, can inhibit the activity of E.coli. It is also possible to produce lactoferrin nanoparticles by electrostatically chelating gellan gum [87], thereby enhancing their antimicrobial capacity without requiring inorganic compounds. This greatly expands the utilization of lactoferrin as a natural antimicrobial agent in food.

There are various types of microorganisms, and microbial-derived agents that reduce bacteria have gradually become the focus of research. Bacteriocins are typically produced by both Gram-positive and Gram-negative bacteria [2]. Studies have shown that lactic acid bacteria isolated from buffalo milk curds produce bacteriocin with antibacterial properties, which can inhibit the growth of pathogenic microorganisms in chickens [88]. Another bacteriocin commonly used in meat is nisin, which is produced by certain strains of lactococcus lactis and has the ability to inhibit various gram-positive bacteria [89]. Arief et al. [90] achieved significant reduction in the growth of Escherichia coli in mutton intestines by adding probiotic lactobacillus plantarum IIA-2C12, which also preserved the color of meat products. However, this addition also resulted in changes to the taste of the meat and decreased acceptability to some extent. Therefore, investigating the impact of microbial preservatives on the flavor of meat products is an important area for future research. Araujo et al. [91] also investigated the bacterioreducing effects of a combination of garlic essential oil (GO), allyl isothiocyanate (AITC), and nisin (NI) on fresh intestines. The combination of 20mg/kg NI + 125 μL/kg GO + 62.5 μL/kg AITC or 20 mg/kg NI + 62.5 μL/kg GO + 125 μL/kg AITC was found to have a significant inhibitory effect on E. coli O157H7 and spoilage lactic acid bacteria, which was better than that of single applications, without affecting the sensory acceptability of fresh intestine.

[2].   Woraprayote, W.; Malila, Y.; Sorapukdee, S.; Swetwiwathana, A.; Benjakul, S.; Visessanguan, W. Bacteriocins from Lactic Acid Bacteria and Their Applications in Meat and Meat Products. Meat Science 2016, 120, 118–132, doi:10.1016/j.meatsci.2016.04.004.

[82]. Fratianni, F.; De Martino, L.; Melone, A.; De Feo, V.; Coppola, R.; Nazzaro, F. Preservation of Chicken Breast Meat Treated with Thyme and Balm Essential Oils. Journal of Food Science 2010, 75, doi:10.1111/j.1750-3841.2010.01791.x.

[83]. Noshad, M.; Behbahani, B.A.; Jooyandeh, H.; Rahmati-Joneidabad, M.; Kaykha, M.E.H.; Sheikhjan, M.G. Utilization of Plantago Major Seed Mucilage Containing Citrus Limon Essential Oil as an Edible Coating to Improve Shelf‐life of Buffalo Meat under Refrigeration Conditions. 2021. Food Science & Nutrition 2021, 19;9(3):1625-1639. doi: 10.1002/fsn3.2137. PMID: 33747474; PMCID: PMC7958549.

[84]. Wang, Y.; Cen, C.; Chen, J.; Zhou, C.; Fu, L. Nano-Emulsification Improves Physical Properties and Bioactivities of Litsea Cubeba Essential Oil. LWT 2021, 137, 110361, doi:10.1016/j.lwt.2020.110361.

[85]. Darmadji, P.; Izumimoto, M. Effect of Chitosan in Meat Preservation. Meat Science 1994, 38 (1994) 243-254.

[86] Ashraf, M.F.; Zubair, D.; Bashir, M.N.; Alagawany, M.; Ahmed, S.; Shah, Q.A.; Buzdar, J.A.; Arain, M.A. Nutraceutical and Health-Promoting Potential of Lactoferrin, an Iron-Binding Protein in Human and Animal: Current Knowledge. Biological Trace Element Research 2024, 202, 56–72, doi:10.1007/s12011-023-03658-4.

[87]. Duarte, L.G.R.; Alencar, W.M.P.; Iacuzio, R.; Silva, N.C.C.; Picone, C.S.F. Synthesis, Characterization and Application of Antibacterial Lactoferrin Nanoparticles. Current Research in Food Science 2022, 5, 642–652, doi:10.1016/j.crfs.2022.03.009.

[88]. Yuliana, T.; Hayati, F.; Cahyana, Y.; Rialita, T.; Mardawati, E.; Harahap, B.M.; Safitri, R. Indigenous Bacteriocin of Lactic Acid Bacteria from “Dadih” a Fermented Buffalo Milk from West Sumatra, Indonesia as Chicken Meat Preservative. Pakistan J. of Biological Sciences 2020, 23, 1572–1580, doi:10.3923/pjbs.2020.1572.1580.

[89]. Dong, A.; Malo, A.; Leong, M.; Ho, V.T.T.; Turner, M.S. Control of Listeria Monocytogenes on Ready-to-Eat Ham and Fresh Cut Iceberg Lettuce Using a Nisin Containing Lactococcus Lactis Fermentate. Food Control 2021, 119, 107420, doi:10.1016/j.foodcont.2020.107420.

[90]. Arief, I.I.; Wulandari, Z.; Aditia, E.L.; Baihaqi, M.; Noraimah; Hendrawan Physicochemical and Microbiological Properties of Fermented Lamb Sausages Using Probiotic Lactobacillus Plantarum IIA-2C12 as Starter Culture. Procedia Environmental Sciences 2014, 20, 352–356, doi:10.1016/j.proenv.2014.03.044.

[91]. Araújo, M.K.; Gumiela, A.M.; Bordin, K.; Luciano, F.B.; Macedo, R.E.F. de Combination of Garlic Essential Oil, Allyl Isothiocyanate, and Nisin Z as Bio-Preservatives in Fresh Sausage. Meat Science 2018, 143, 177–183, doi:10.1016/j.meatsci.2018.05.002.

Q40: Verify the text of Chapter 4.4. Is it about "hurdle technology"?

A40: Thanks for your reminding. You are right, it is hurdle technology. In the revised manuscript, this part was supplemented and rewritted in line 568-584 as follow:

3.4. Application of hurdle technology in Meat

The above bacteriological reduction technologie can be used to improve the safety and shelf life of food ingredients. However, the antibacterial effects of some of these technologies are not obvious when used alone and adversely affect the organoleptic properties of foods and reduce consumer acceptability. Owing to these facts, the hurdle concept (generally known as combined methods, combination preservation, combined processes, barrier technology or combination techniques) has become a promising technology that simultaneously reduces losses of nutritional and sensory quality and improves food safety [41]. The hurdle technology is a method for ensuring the safety of food products by controlling or destroying microbial spoilage or food pathogens through the combination of a series of preservative parameters (hurdles) imposing physiological impact on microbial cells. Early studies have shown that the preservation period of foods with low initial bacterial counts can be extended 1 ~ 2 times longer than those with high initial bacterial counts. If the initial amount of bacteria inside the food is low, only a few defense factors are needed for effective bacterial inhibition; conversely, if poor sanitary conditions cause a high initial amount of bacteria, it is necessary to increase the defense factor or enhance its strength.

Q41: Chapter 4 contains only general information. It is necessary to supplement the content of the chapter, for example, with parameters for conducting an effective process. It should be completed with HPP technology, too.

A41: Thanks for your reminding. In the revised manuscript, this chapter has been updated to include parameters for valid processes and HPP in line 255-567.

3.1. Application of chemical bacteria reduction technology in meat

3.1.1. SAEW

The bactericidal effect of SAEW varies slightly depending on the treatment method used, such as soaking, atomization, spraying, washing, etc. Among these methods, soaking treatment yields the best bactericidal effect. Rahman et al. [41] soaked fresh chicken breast with SAEW, which significantly reduced the number of natural flora and pathogenic bacteria, and slowed down the rate of spoilage during refrigeration. Similarly, Sheng et al. [42] discovered that soaking beef in SAEW was effective in killing microorganisms and delaying the deterioration of refrigerated beef quality. Furthermore, the bactericidal effect of SAEW varies depending on the duration and concentration of treatment. Ye et al. [43] investigated the effects of different treatment times and concentrations of SAEW on raw frozen shrimp and found that treating with SAEW for 5 minutes had a better bactericidal effect than treating with SAEW for 3 minutes. Additionally, under the same treatment time, the bactericidal effect of 29 mg/L of SAEW was significantly enhanced compared to that of 16 mg/L of SAEW. However, SAEW has limited ability to maintain oxidative stability during meat storage. Therefore, combining SAEW with antioxidants can help enhance its oxidative stability. However, further research is needed. In recent years, SAEW ice has been studied and applied for the preservation of meat and aquatic products [44]. The storage and preservation of aquatic products with ice containing bactericidal substances can not only inhibit the reproduction of microorganisms on the surface of the products but also extend their shelf life.

Currently, although SAEW technology has been widely utilized in the domestic and international food industry, there is still a lack of systematic research and evaluation on the sterilization principle of SAEW, influencing factors, and its impact on the oxidation of different types of meat. This deficiency also hinders the promotion and application of SAEW in meat processing and industrialization. Maximizing the sterilization and preservation effect of SAEW, as well as exploring its combination with other technologies, remains the focus of future research.

3.1.2. Organic acids

The bacterial reduction effect of organic acids is closely related to their species, concentration, pH, volume, the type of target microorganisms, and temperature of the carcass surface. Organic acids are often sprayed on meat to reduce bacteria. However, soaking the meat in organic acids can lead to cross-contamination and significant loss of soluble substances, which affects the effectiveness of bacteria reduction. Some scholars have explored the bacteria-reducing effect of nine different organic acids (including lactic acid, acetic acid, citric acid, peracetic acid, etc.) on cattle carcasses and found that lactic acid is the most effective measure for reducing bacteria. It can effectively reduce the total number of colonies and coliform bacteria [45]. In addition, lactic acid is the most commonly used organic acid in the meat industry to reduce bacteria in products due to its effectiveness and cost. Furthermore, the European Union permits the spraying of 2 % to 5 % lactic acid on carcasses as a means of reducing bacteria [46]. Ransom et al. [47]demonstrated that applying 2 % lactic acid can decrease E. coli O157:H7 on cattle carcass surfaces by 1.6 Log (CFU/g). Manzoor et al. [48] also compared the effects of different concentrations of lactic acid spray on the microbial and sensory characteristics of buffalo meat and found that both could significantly reduce the number of microorganisms, while a concentration of lactic acid above 6 % would adversely affect the color of the meat. When a higher temperature (above 55 ℃) is combined with a lower concentration (2 %) of lactic acid spray, the bactericidal effect is enhanced. However, the stability of organic acids is compromised under high temperature conditions, leading to changes in meat's sensory properties and the emergence of acid-resistant pathogens. Additionally, this may also result in corrosion to processing equipment [45]. The study found that the combination of two or more organic acids is more effective than a single organic acid, which can enhance the bactericidal effect and overall food quality. Surve et al. [49] found that combining acetic acid with lactic acid or propionic acid can significantly enhance the antimicrobial properties of buffalo meat without affecting its color and flavor, effectively extending its refrigerated shelf life.

In the future, new organic acids or derivatives of organic acids could be developed to address the issues of heat resistance and broad-spectrum antimicrobial properties associated with existing organic acids. Despite some limitations in food sterilization, organic acids have broad application prospects as a natural and environmentally friendly sterilization technology due to increasing consumer demand for natural and safe foods.

3.1.3. Ozone

The application of ozone in meat can be divided into two forms: gaseous state and aqueous solution[12]. Its safety and effectiveness have been proven for killing a variety of foodborne pathogens in food, such as Escherichia coli, Salmonella, and Listeria monocytogenes [50]. Cárdenas et al. [51] analyzed the antibacterial effect of gaseous ozone on chilled fresh beef and found that an ozone concentration of 141.12 mg/m3 could reduce the total number of E. coli and inoculated microorganisms, resulting in complete inactivation of certain microorganisms and maintaining meat freshness. The bactericidal effect of ozone is influenced by factors such as concentration, temperature, contact time, and food characteristics. Stivarius et al. [52] investigated the bactericidal effect of a 1% ozone aqueous solution and found that a 7-minute ozone treatment had a better bactericidal effect than a 5-minute treatment. Additionally, the researchers compared high ozone dose (1000 ppm) with low ozone dose (100 ppm) and discovered that although the high ozone dose can slow down microbial activity on the surface of pork and reduce physiological activity, it is not sufficient to have a fatal effect on microbial life in meat. This may be attributed to the longer incubation time (46 and 49 hours) of samples under non-sterile conditions after ozone treatment [53]. However, ozone is also a potent oxidizing agent that can impact the sensory, physical, and chemical properties of meat and meat products. Ayranci et al. [54] discovered that ozone treatment (1 x 10-2 kg m-3, 8 hours) had detrimental effects on the physicochemical properties and color of turkey breast. The high oxidation potential of ozone may be the primary reason for lipid and protein oxidation in meat. Additionally, it was observed that the connective tissue membrane with a high protein content in turkey meat underwent denaturation after ozone treatment, which could explain the lighter coloration of turkey meat.

Ozone sterilization systems require complex equipment and precise control, making them costly to operate and maintain. Furthermore, high concentrations of ozone can be toxic to the human body, so it is necessary to strictly regulate the concentration and contact time during actual operation in order to ensure food safety and protect operators.

3.2. Application of non-thermal physical bacteria reduction technology in meat

3.2.1. Ultrasound

In the food industry, high-power ultrasonic waves with a frequency of 20 ~ 100 kHz are generally used to inactivate microorganisms [55]. The sterilization effect of ultrasonication is mainly influenced by the ultrasonic medium, frequency, action time, and microbial species [56]. Huu, et al. [57] used ultrasonic treatments with a frequency of 40 kHz and a power density of 0.092 W/mL for E. coli O157:H7 or L. innocua. The initial bacterial content was 1 × 106 CFU/mL, and the bacterial content remained at the initial level after both 30 min and 45 min of ultrasonic treatment. The results showed that there was no significant influence on either bacteria in the product during the entire period of ultrasonic treatment. It has been found that the combination of ultrasonic and heat treatment can accelerate the sterilization speed of food. Morild et al. [58] evaluated the effect of pressurized steam combined with high-power ultrasound on pathogen inactivation on pig skin and pork surface. The study examined the inactivation of Salmonella typhimurium, Salmonella Derby, Salmonella infantile, Yersinia enterocolitica, and a non-pathogenic E. coli. After 4 seconds of ultrasound treatment, the total number of colonies decreased by 3.3 log CFU/cm2. Musavian et al. [59] also found that steam treatment and ultrasonic treatment of chicken carcasses on the processing line could significantly reduce the number of campylobacter on contaminated poultry. By using steam and ultrasound immediately after slaughter, the total count of viable bacteria was reduced by approximately 3 log CFU/cm2. However, prolonged exposure of food to high temperatures at ultrasonic wavelengths leads to a reduction in its functional properties, sensory properties, and nutritional value [60]. Therefore, it can be combined with other technologies that reduce bacteria to enhance its bactericidal effect. Kordowska-Wiater et al. [61] investigated the isolation of Salmonella, Escherichia coli, Proteus, and Pseudomonas fluorescence from the surface of chicken skin after being treated with ultrasound (at a frequency of 40 kHz, intensity of 2.5 W cm2, and duration of either 3 or 6 minutes) in water and a 1 % lactic acid solution. It was observed that treating the chicken skin with ultrasound alone in a lactic acid solution for 3 minutes resulted in a reduction of 1.0 log CFU/cm2 in the number of these microorganisms. Furthermore, extending the treatment time to 6 minutes led to a decrease exceeding 1.0 log CFU/cm2 in the number of microorganisms present in the water sample.

Ultrasound have varying effects on the inactivation of different microorganisms (bacteria, viruses, fungi, and mycotoxins) as well as food substrates. The frequency, intensity, and treatment time of ultrasonic waves need to be optimized for different types of food. Several studies have shown that while ultrasound can reduce the number of bacteria produced by meat, using two or more bacteria-reducing technologies is more effective than relying solely on bacterial reduction [21]. Therefore, it is rarely used alone in actual meat production processes for reducing bacteria and usually works in conjunction with other bacteria-reducing technologies.

3.2.2. Irradiation

Irradiation technology has been approved by the Food and Drug Administration (FDA) for use in food processing. Food irradiation is a form of "cold treatment" that effectively kills bacteria without significantly increasing the internal temperature or causing nutrient loss in the food [47]. Electron beam radiation (EB) and X-ray radiation (XR) are more acceptable to consumers than gamma rays (GR) because they do not involve radioactive isotopes [48]. However, Park et al. [62] found that GR irradiation may be more effective in reducing bacterial populations than EB irradiation. Additionally, doses of 5~10 kGy for GR and EB irradiation respectively had no adverse effects on the lipid oxidation and sensory characteristics (color, chewability, and taste) of beef sausage patties. The technical advantages of high-energy X-rays include better power utilization, dose uniformity, and shorter irradiation times. Consequently, the use of X-ray irradiation results in higher productivity and lower processing costs. Yim et al. [63] used X-ray irradiation to treat beef samples, and as the irradiation dose increased, there was a significant decrease in the total number of aerobic bacteria present. No bacterial growth was observed in samples treated with a dose of 10 kGy. The combined treatment of irradiation and natural antimicrobial agents in raw meat materials can enhance the degree of inhibition against food-borne pathogens and extend the shelf life of meat and meat products. Hu et al. [64] combined chitosan-eugenol with irradiation to significantly reduce the number of Staphylococcus aureus and Salmonella in fresh pork, as well as delay fat oxidation during processing.

However, although the meat and its products treated with low-dose radiation will not be contaminated by radioactivity and play a good role in inhibiting the growth of pathogenic microorganisms in meat, it is important to select the appropriate radiation for each type of food during use. Additionally, parameters such as radiation dose and time should be well controlled to minimize adverse effects on the quality of meat flavor, color, nutrition, and moisture [65].

3.2.3. Ultraviolet

UV sterilization technology has the advantages of a broad spectrum, high efficiency, and no secondary pollution. It has been proven to effectively reduce pathogenic microorganisms on the surface of meat products and extend their shelf life. It is widely used in meat preservation. In the study conducted by Söbeli et al. [66], various doses of UV were used to irradiate beef fillet, and it was observed that a higher dose (4.2 J/cm2) significantly reduced the total number of aerobic bacteria by 3.49 ± 0.67 log CFU/g. In contrast, Bryant et al. [67] investigated the effect of UV radiation on the inactivation of E. coli K12 on beef surfaces. It was shown that the microbial inactivation by UV treatment was correlated with both the duration of treatment and the distance between the sample and the light source. Generally, the longer the treatment time and the shorter the distance between UV and the sample, the higher will be the rate of microbial inactivation. The bactericidal effect of ultraviolet light varies depending on the specific type of microorganism. McLeod et al. [55] investigated the inactivation of pathogenic bacteria in chickens treated with continuous UVC. The results demonstrated that exposure to 30 J/cm2 of UVC reduced the log CFU/g counts of C. divergens, enterohemorrhagic E. coli and pseudomonas spp. by 2.8, 1.7, and 2.7 log CFU/g respectively. However, due to the poor penetration ability of UV in reaching the interior of the food, it can only reduce the survival rate of the initial bacterial count on the surface of the meat sample and cannot eliminate all microorganisms. Therefore, it is often used in combination with other sterilization technologies to enhance its effectiveness. Studies have found that the use of UV in combination with ozone [68] or other methods, such as peracetic acid and lactic acid [69], can effectively eliminate microorganisms in meat. At the same time, meat may undergo a series of oxidation processes after being exposed to a high dose of UV treatment, which can potentially affect the quality of the product. To mitigate the UV oxidation of meat lipids and proteins, hurdle technology can be employed. For example, UV can be combined with deaerator packaging, vacuum packaging, and high hydrostatic pressure [70,71]. Therefore, the development of diverse UV combination technologies is essential to ensure the quality and safety of meat in future UV applications.

Compared to fruits and vegetables, UV is used in meat and meat products to a relatively small extent. The bactericidal effect of different UV irradiation doses on meat varies. However, there is currently limited research on the relationship between dose and meat disinfection. Additionally, the application of ultraviolet in meat needs further exploration regarding appropriate wavelength, dose, and luminescence methods for different types of meat. Therefore, future work should focus on expanding basic research on UV in meat and meat products as well as researching and developing new UV technologies.

3.2.4. Cold plasma

In recent years, Cold plasma has gained widespread attention in the field of food sterilization and shows great potential in meat preservation applications. It can effectively eliminate various types of viruses, bacteria, fungi, and spores. The ability of Cold plasma to inhibit bacteria mainly depends on factors such as gas composition, airflow, electrical input, and duration of the process[72]. Kim et al. [73] used helium (10 lpm) and a mixture of helium and oxygen (10 lpm and 10 sccm) as the excitation medium to process sliced bacon under the conditions of input power at 75, 100, and 125 W, with treatment times of 60 and 90 seconds respectively. After plasma helium treatment and helium/oxygen mixed treatment, the total number of aerobic bacteria decreased by 1.89 decimals and 4.58 decimals respectively. The study also found that regardless of gas composition, the bacteria-reducing effect is enhanced with increasing input power and treatment time. In addition, low-temperature plasma has different effects on reducing bacteria in different microbial species. Choi et al. [74] used a 20 kV DC with an output voltage of 58 kHz frequency. It was found that the optimal condition for microorganism inactivation was achieved when the plasma was generated at a current of 1.5 A and the distance between the plasma electrode tip and the sample was 25 mm. After treating pork samples for 120 seconds, it was observed that E.coli O157:H7 had a stronger inactivation effect compared to Lactobacillus monocytogenes, and there were no significant changes in pork color, flavor, nutrition, or other quality indexes after treatment. Moreover, Ulbin-Figlewicz et al. [75] conducted a study on the treatment of helium and argon plasma to compare their effects on inactivating microflora on the surfaces of different types of meat. Under an operating pressure of 20 kPa, after 10 minutes of helium plasma treatment, the total number of colonies in pork decreased by 1.14 to 1.48 logarithmic cycles, while that in beef only decreased by 0.98 to 2.09 logarithmic cycles.

Currently, technology plays a significant role in the sterilization and preservation of meat and meat products; however, it is still in the early stages of basic research with some issues, such as weak penetration ability. When a large number of microorganisms gather on the food surface, resulting in uneven bactericidal effects, its effectiveness will be affected. The technology can be combined with other non-heat treatment methods to enhance its bactericidal effect. Additionally, there are challenges like high equipment investment costs. The specific influencing factors, mechanism of action, and regulatory means of this technology lack a well-established theoretical foundation for support and require further exploration in future studies to promote wider application in the meat industry.

3.2.5. HPP

The effectiveness of HPP for microbial control depends on various factors including food composition, types of microbes present, the method and magnitude of applied pressure, duration of pressure application, treatment temperature as well as the integrity of packaging [76]. Hayman et al. [77] conducted a study using HPP on different types of meat (low-fat pastrami, Strasbourg beef, export sausage, and Cajun beef), which demonstrated that treatment at 600 MPa and 20℃ for 180 seconds extended the refrigerated shelf life of ready-made meats. The amount of L. monocytogenes in the inoculated products was reduced by more than 4 log CFU/g. In order to enhance the bactericidal effect of HPP, antibacterial agents can be combined with HPP. Melhem et al. [78] investigated the combined inactivation effect of HHP on pathogenic and spoilage bacteria in meat products using different application methods (surface application, product incorporation, and active packaging) and various types of antibacterials (derived from plants, microorganisms, and animals). It was discovered that combining bacteriocin with HHP resulted in a stronger bacteriocidal effect due to its significant inhibitory impact on gram-positive bacteria like listeria. Regarding application methods, the synergistic effect of active packaging and HHP proved to be the most effective approach for pathogenic bacteria inactivation in meat products. However, the use of HPP may have an effect on the color of fresh meat, and the exact mechanism of color change is unknown. This could be attributed to the disruption of non-covalent bonds, protein denaturation, and myoglobin oxidation in fresh meat under high pressure [76,79]. Studies have shown that compared with beef or mutton treated with HPP, poultry meat has lower myoglobin content and is less affected by HPP treatment [80]. To investigate this issue, Gupta et al. [79] discovered that altering the myoglobin status can reduce the discoloration caused by HPP. Patties were prepared using different packaging methods (high oxygen - oxymyoglobin, carbon monoxide - carboxymyoglobin) or through the addition of potassium ferricyanide (metmyoglobin). It was found that carboxymyoglobin exhibited better color retention.

The application of HPP as a non-thermal physical technology in meat bacterialization has shown great potential, but it still faces some challenges. For instance, bacterial spores are highly resistant to high pressure [81], requiring higher pressures (> 1200 MPa) [28] for their inactivation. However, these higher pressure conditions may have adverse effects on food quality. Therefore, in the future, it can be used in combination with other methods to reduce bacteria and sterilize food.

3.3. Application of biological bacteria reduction technology in meat

Natural antimicrobials have long been used as food additives, primarily to extend the shelf life and maintain the quality of meat and meat products. Among natural antibacterial compounds, plant-derived antimicrobials have been extensively studied, and substances such as phenols, isoflavones, ketones, acids, terpenes (essential oils), and alkaloids commonly found in meat can limit or hinder the growth of harmful microorganisms. Different types of microorganisms in meat and meat products are inhibited by various plant extracts. Studies have shown that when thyme oil and balsam oil [82] are applied to chicken breast, balsam oil significantly restricts the growth of salmonella while thyme oil effectively inhibits the growth of E. coli. Furthermore, they are incorporated into meat in various forms, such as direct mixing, coating, layering, nanoencapsulation, and microencapsulation. This can enhance its antibacterial properties and significantly expand the application of natural antimicrobials in meat. For instance, Noshad et al. [83] blended lemon essential oil with psyllium seed slime to create a novel edible coating for beef storage and preservation, and discovered that it could extend the shelf life by 10 days. Moreover, different addition technologies may have varying inhibitory effects on microorganisms. This was demonstrated by Wang et al. [84], who found that nanoemulsions of litsea cubeba essential oil exhibited stronger inhibition against Listeria monocytogenes and Streptococcus maritimus compared to pure litsea cubeba essential oil.

Chitosan, a polycationic biopolymer commonly extracted from the exoskeletons of crustaceans such as crabs and lobsters, is frequently used as an animal-derived antimicrobial agent in meat. Darmadji et al. [85] discovered that chitosan at concentrations ranging from 0.1 % to 1% effectively inhibits the growth of spoilage bacteria and pathogenic bacteria during the storage of refrigerated meat. In addition to chitosan, lysozyme extracted from eggs and milk is also commonly employed. Furthermore, lactoferrin [86], a natural antibacterial substance found in secretions like saliva, milk, and tears of mammals, can inhibit the activity of E.coli. It is also possible to produce lactoferrin nanoparticles by electrostatically chelating gellan gum [87], thereby enhancing their antimicrobial capacity without requiring inorganic compounds. This greatly expands the utilization of lactoferrin as a natural antimicrobial agent in food.

There are various types of microorganisms, and microbial-derived agents that reduce bacteria have gradually become the focus of research. Bacteriocins are typically produced by both Gram-positive and Gram-negative bacteria [2]. Studies have shown that lactic acid bacteria isolated from buffalo milk curds produce bacteriocin with antibacterial properties, which can inhibit the growth of pathogenic microorganisms in chickens [88]. Another bacteriocin commonly used in meat is nisin, which is produced by certain strains of lactococcus lactis and has the ability to inhibit various gram-positive bacteria [89]. Arief et al. [90] achieved significant reduction in the growth of Escherichia coli in mutton intestines by adding probiotic lactobacillus plantarum IIA-2C12, which also preserved the color of meat products. However, this addition also resulted in changes to the taste of the meat and decreased acceptability to some extent. Therefore, investigating the impact of microbial preservatives on the flavor of meat products is an important area for future research. Araujo et al. [91] also investigated the bacterioreducing effects of a combination of garlic essential oil (GO), allyl isothiocyanate (AITC), and nisin (NI) on fresh intestines. The combination of 20 mg/kg NI + 125 μL/kg GO + 62.5 μL/kg AITC or 20 mg/kg NI + 62.5 μL/kg GO + 125 μL/kg AITC was found to have a significant inhibitory effect on E. coli O157H7 and spoilage lactic acid bacteria, which was better than that of single applications, without affecting the sensory acceptability of fresh intestine.

However, in practical application, the development and utilization of natural antimicrobial agents are still imperfect, with problems such as instability, insufficient extraction of available active substances, and limited effectiveness of a single antimicrobial agent, which may prevent achieving the expected results. To better utilize natural antimicrobials, new composite antimicrobials can be synthesized by optimizing their extraction and purification methods and utilizing new technologies [84] to enhance their stability and antibacterial effects. This will facilitate their improved application in the field of meat preservation.

Q42: Conclusions should be rewritted bocause the title and aim of manuscript.

A42: Thanks for your reminding. In the revised manuscript, the sentence has been corrected in line 602-622 as follow:

This review summarizes the characteristics, bactericidal principles, and applications of commonly used bactericidal technologies in meat. Each individual bactericidal technology has its own characteristics and can achieve the expected effect to a certain extent. However, there are still limitations as the bactericidal effect is easily restricted by different objective factors. Chemical bactericidal technology is widely used due to its high efficiency; however, frequent use of the same chemical agent may lead to microbial resistance. Non-thermal physical bacteria reduction technology causes less damage to food color, flavor, and nutrition. It does not cause pollution. However, it typically requires a specific environment and equipment, resulting in high energy consumption. Additionally, the killing effect on microorganisms may be unstable, especially when there are significant changes in environmental conditions or complex microbial species present. As a result, this can affect the overall effectiveness and comprehensiveness of the technology. Biological bacteria reduction technology is safe, utilizes a wide range of biological resources, exhibits high levels of biodiversity and adaptability. However, it is easily influenced by extraction rate, climate, geographical conditions, and processing costs. With continuous development and in-depth research on bactericidal technology based on the "hurdle technology" principle, different bactericidal pretreatment measures are combined according to the type of food. This targets microorganisms in meat raw materials for control purposes, thereby enhancing meat quality and ensuring food safety. These advancements will shape the future trend of meat preservation while facilitating its industrialization and application.

Q43: Mismatches between references and bibliography should be removed. The literature list contains many errors (journal names abbreviated and full names - e.g., 1 and 4, no authors - e.g., 22, no data - e.g., 51). It should be carefully reviewed for compliance with editorial requirements.

A43: Thanks for your reminding. In the revised manuscript, the sentence has been corrected in line 635-875.

Round 2

Reviewer 1 Report

Comments and Suggestions for Authors

Authors improved significantly the manuscript and I recommend it for publication.

Author Response

Thanks for your approval of our revised manuscript.

Reviewer 3 Report

Comments and Suggestions for Authors

In my opinion, the substantive quality of the manuscript has increased. However, the authors keep making editorial mistakes, repeating the same information in different parts of the manuscript (e.g., characterizing the methods), but omitting the method of literatura research.

I believe that the authors' explanation marked A09 should be included in the manuscript as a description of the literature survey method. Preferably before chapter 2. In fact, it is a description of the literature research method used. Since different authors may use different methods and obtain different results this description is necessary in my opinion.

Detailed comments:

l.92 - unnecessary space before COOH;

l.111 - dot expected at the end of sentence;

l.124 - unnecessary double space after dot;

l.129 - rather "kHz";

Table 2 - HClO and ClO (lowercase "l") should be written instead of HCIO and CIO;

Table 1 and Figure 1 - the used acronym/abbreviation/initialism should be clarified according to the "Instructions for Authors" (It should be defined the first time they appear in each of three sections: the abstract; the main text; the first figure or table.);

l.295 - space needed before "demonstrated";

l.296,350... - "O157:H7" should not be italic;

l.358 - serovar "Derby" should not be italic;

l.361 - "Campylobacter"

l.296-297 compare with l.363,373-374,415,454, 468-469... - standardize the spelling;

l.368 - verify the intensity unit;

l.424 - "enterohemorrhagic" should not be italic;

l.424 - rather "Pseudomonas", and spp. should not be italic;

l.446,448... - rather "cold" (lowercase "c");

l.450 - necessary space after "process";

l.485 compare with l.301 - standarize degree symbol;

l.519-530 - do not use italic in "thyme oil", "balsam oil", "lemon essential oil", but italic should be used in latin names "Salmonella" (uppercase "S"), "Litsea cubeba" (uppercase "L");

l.538 - necessary space before "coli";

l.544-545 - "lactic acid bacteria" is not latin name - not italic; but...

l.547 - "Lactococcus lactis" (uppercase "L"); and...

l.550 - "Lactobacillus plantarum";

l.548 - "Gram-" (uppercase "G");

l.548-559 - do not use italic in "probiotic", "IIA-2C12", "garlic essential... nisin (NI)", "spoilage lactic acid bacteria";

l.568 - "meat" (lowercase "m");

l.588 - unnecesary space after "š";

References should be described as pointed in "Instructions for Authors". Especially, abbreviated journal name should be used. The Authors need to verify this thoroughly.

Author Response

Dear Editors and Reviewers:

Thank you for your letter and for the reviewers’ comments concerning our manuscript entitled “Research Progress on Bacteria-Reducing Pretreatment Technology of Meat”. Thanks to the experts for reviewing our article and giving us honest advice to improve the manuscript. Thank you very much for giving us the opportunity to rework our work and improve the manuscript. You are right that much revision and clarification need to be done in order to paper might possibly be published. Thanks again.

The manuscript has been revised carefully according to your and the reviewers' comments. In our rebuttal indicate the line number in the revised manuscript corresponding to each change that has been made and use yellow highlighting in the text to indicate the edits. We have carefully made correction which we hope to meet with approval. Changes are marked in red in the manuscript. The responds to the reviewers comments and all corrections in the paper are as follow:

Reviewer 3

Thank you very much for the recognition of this study. We would like to express our most sincere appreciation for your careful review of this manuscript. The valuable suggestions you had proposed were beneficial to improve the quality of this manuscript. The modifications have been performed carefully according to your suggestions as follow:

Q1: In my opinion, the substantive quality of the manuscript has increased. However, the authors keep making editorial mistakes, repeating the same information in different parts of the manuscript (e.g., characterizing the methods), but omitting the method of literatura research.

A1: Thanks for your suggestion. In the revised manuscript, as the characteristics of different technologies are described in Table 1, we have removed duplicate content. The specific modifications are as follows:

  • It possesses characteristics such as efficient sterilization, convenient manufacturing, low cost, wide application, safety and environmental friendliness. (line 85-86)
  • The technology possesses characteristics such as low cost, green safety, wide applicability, and convenience. (line 136-138)
  • However, it is a highly efficient method for sterilization, cold treatment with no residue, strong controllability, and environmentally friendly food treatment that saves energy. (155-157)
  • It is considered a broad-spectrum, efficient, environmentally friendly, and residue-free technology for reducing bacteria. (line 169-170)
  • However, cold plasma sterilization technology has the characteristics of being mild, highly efficient, and residue-free, which will play an important role in optimizing the meat preservation process. (line 183-185)
  • Despite the high equipment cost of HPP in the production process, especially in large-scale applications, which results in a significant investment that may affect the texture of food, it offers advantages such as short processing time, low energy consumption, uniform pressure effect, and green safety.(line 194-197)
  • With the characteristics of wide source, low cost, environmental friendliness, broad-spectrum antimicrobial, and no resistance to the target of action, they meet the requirements of sustainable development strategies and are widely used in meat preservation.(line 203-206)
  • However, there is obvious seasonality and regionalism, and the separation and extraction process are subject to many restrictions.(line 214-215)
  • It is a class of bacteriostatic substances with natural origin, broad-spectrum, safety, biocompatibility and easy application, which is of great significance to improve the safety and prolong the shelf life of food.(line 222-225)
  • It is widely available and safe, but the metabolites of different types of microorganisms vary greatly in terms of antibacterial ability and range, which imposes certain limitations on its practical application.(line 232-235)

Q2: I believe that the authors' explanation marked A09 should be included in the manuscript as a description of the literature survey method. Preferably before chapter 2. In fact, it is a description of the literature research method used. Since different authors may use different methods and obtain different results this description is necessary in my opinion.

A2: Thanks for your suggestion. In the revised manuscript, the literature survey method has been supplemented in line 55-61 as follow:

The databases such as web of science, scopus, pubmed, and science direct were applied to systematically search literature. The search methods include keyword search (bacteria reduction technology, bacteria/non-thermal physical/reduction technology, organic acid, ozone, ultrasonic etc.), journal search (foods, food chemistry, meat science etc.), and citation literature search (based on the situation of target article cited by other articles). Previous studies have extensively examined the application of single bacteriological reduction technology in food [10–12].

Q3: l.92 - unnecessary space before COOH;

A3: Thanks for your suggestion. In the revised manuscript, the sentence has been corrected in line 91-92 as follow:

Organic acids, as food-grade antibacterial agents, are a class of organic compounds that contain carboxylic (-COOH) functional groups.

Q4: l.111 - dot expected at the end of sentence;

A4: Thanks for your suggestion. In the revised manuscript, the sentence has been corrected in line 109-111 as follow:

It is also easy to decompose into HO- and O2 in water, which has a strong disinfecting and sterilising ability [19].

Q5: l.124 - unnecessary double space after dot;

A5: Thanks for your suggestion. In the revised manuscript, the sentence has been corrected in line 121-124 as follow:

Compared with the traditional thermal sterilization technology, usually at low temperature or room temperature can achieve the purpose of sterilization, can better ensure the quality of food. Commonly used methods for sterilizing meat raw materials include ultrasound, radiation, ultraviolet light, cold plasma, high-pressure processing, etc.

Q6: l.129 - rather "kHz";

A6: Thanks for your suggestion. In the revised manuscript, the sentence has been corrected in line 126-129 as follow:

It can travel through air, liquids, and solids and has the capability to kill certain microorganisms in food. In the food industry, ultrasound at a lower frequency (20 ~ 100 kHz) is usually used to deactivate microorganisms.

Q7: Table 2 - HClO and ClO (lowercase "l") should be written instead of HCIO and CIO;

A7: Thanks for your suggestion. In the revised manuscript, the sentence has been corrected in Table 2 as follow:

The cell membrane is damaged by SAEW, causing rapid leakage of K+ and an increase in membrane permeability. This causes HCIO and CIO- to enter the cell, resulting in the following consequences:

Q8: Table 1 and Figure 1 - the used acronym/abbreviation/initialism should be clarified according to the "Instructions for Authors" (It should be defined the first time they appear in each of three sections: the abstract; the main text; the first figure or table.);

A8: Thanks for your suggestion. In the revised manuscript, the sentence has been corrected in line 13-16 as follow:

This review summarizes commonly used technologies for reducing bacteria in meat, including slightly acidic electrolyzed water (SAEW), organic acids, ozone (O3), ultrasound, irradiation, ultraviolet (UV), cold plasma, high-pressure processing (HPP), and biological bacterial reduction agents.

Q9: l.295 - space needed before "demonstrated";

A9: Thanks for your suggestion. In the revised manuscript, the sentence has been corrected in line 274-276 as follow:

Ransom et al. [47] demonstrated that applying 2 % lactic acid can decrease E. coli O157:H7 on cattle carcass surfaces by 1.6 Log CFU/g.

Q10: l.296,350... - "O157:H7" should not be italic;

A10: Thanks for your suggestion. In the revised manuscript, the sentence has been corrected in line 274-276 and 328-329 as follow:

Ransom et al. [47] demonstrated that applying 2 % lactic acid can decrease E. coli O157:H7 on cattle carcass surfaces by 1.6 Log CFU/g.

Huu et al. [57] used ultrasonic treatments with a frequency of 40 kHz and a power density of 0.092 W/mL for E. coli O157:H7 or L. innocua.

Q11: l.358 - serovar "Derby" should not be italic;

A11: Thanks for your suggestion. In the revised manuscript, the sentence has been corrected in line 336-337 as follow:

The study examined the inactivation of Salmonella Typhimurium, Salmonella Derby, Salmonella Infantile, Yersinia enterocolitica, and a non-pathogenic E. coli.

Q12: l.361 - "Campylobacter"

Q12: Thanks for your suggestion. In the revised manuscript, the sentence has been corrected in line 339-341 as follow:

Musavian et al. [59] also found that steam treatment and ultrasonic treatment of chicken carcasses on the processing line could significantly reduce the number of Campylobacter on contaminated poultry.

Q13: l.296-297 compare with l.363,373-374,415,454, 468-469... - standardize the spelling;

A13: Thank you for your suggestion. Due to the different methods of calculating the total number of colonies in different literatures, the units used are also different. Non-standard units have been adjusted. In the revised manuscript, the sentence has been corrected as follow:

  • Ransom et al. [47] demonstrated that applying 2 % lactic acid can decrease coli O157:H7 on cattle carcass surfaces by 1.6 log CFU/g. (line 274-276)
  • The results demonstrated that exposure to 3.0 J/cm2 of UVC reduced the log CFU/cm2 counts of Carnobacterium divergens, enterohemorrhagic coli and Pseudomonas spp. by 2.8, 1.7, and 2.7 log CFU/cm2 respectively. (line 403-405)
  • After plasma helium treatment and helium/oxygen mixed treatment, the total number of aerobic bacteria decreased by 1.89 and 4.58 log CFU/g (line 433-435)
  • Under an operating pressure of 20 kPa, after 10 minutes of helium plasma treatment, the total number of colonies in pork decreased by 1.14 to 1.48 log CFU/cm2, while that in beef only decreased by 0.98 to 2.09 log CFU/cm2. (line 446-449)

Q14: l.368 - verify the intensity unit;

A14: Thanks for your suggestion. In the revised manuscript, the sentence has been corrected in line 348-349 as follow:

(at a frequency of 40 kHz, intensity of 2.5 W/cm2, and duration of either 3 or 6 minutes)

Q15: l.424 - "enterohemorrhagic" should not be italic; rather "Pseudomonas", and spp. should not be italic;

A15: Thanks for your suggestion. In the revised manuscript, the sentence has been corrected in line 403-405 as follow:

The results demonstrated that exposure to 3.0 J/cm2 of UVC reduced the log CFU/cm2 counts of Carnobacterium divergens, enterohemorrhagic E. coli and Pseudomonas spp. by 2.8, 1.7, and 2.7 log CFU/cm2 respectively.

Q16: l.446,448... - rather "cold" (lowercase "c");

A16: Thanks for your suggestion. In the revised manuscript, the sentence has been corrected in line 426-430 as follow:

In recent years, cold plasma has gained widespread attention in the field of food sterilization and shows great potential in meat preservation applications. It can effectively eliminate various types of viruses, bacteria, fungi, and spores. The ability of cold plasma to inhibit bacteria mainly depends on factors such as gas composition, airflow, electrical input, and duration of the process [73].

Q17: l.450 - necessary space after "process";

A17: Thanks for your suggestion. In the revised manuscript, the sentence has been corrected in line 428-430 as follow:

The ability of cold plasma to inhibit bacteria mainly depends on factors such as gas composition, airflow, electrical input, and duration of the process [73].

Q18: l.485 compare with l.301 - standarize degree symbol;

A18: Thanks for your suggestion. In the revised manuscript, the sentence has been corrected in line 279-281 and 463-466 as follow:

When a higher temperature (above 55 °C) is combined with a lower concentration (2 %) of lactic acid spray, the bactericidal effect is enhanced.

Hayman et al. [78] conducted a study using HPP on different types of meat (low-fat pastrami, Strasbourg beef, export sausage, and Cajun beef), which demonstrated that treatment at 600 MPa and 20 °C for 180 seconds extended the refrigerated shelf life of ready-made meats.

Q19: l.519-530 - do not use italic in "thyme oil", "balsam oil", "lemon essential oil", but italic should be used in latin names "Salmonella" (uppercase "S"), "Litsea cubeba" (uppercase "L");

A19: Thanks for your suggestion. In the revised manuscript, the sentence has been corrected in line 499-510 as follow:

Studies have shown that when thyme and balm essential oils [83] are applied to chicken breast, balm essential oil significantly restricts the growth of Salmonella sp. while thyme essential oil effectively inhibits the growth of E. coli. Furthermore, they are incorporated into meat in various forms, such as direct mixing, coating, layering, nanoencapsulation, and microencapsulation. This can enhance its antibacterial properties and significantly expand the application of natural antimicrobials in meat. For instance, Noshad et al. [84] blended Citrus limon essential oil with psyllium seed slime to create a novel edible coating for beef storage and preservation, and discovered that it could extend the shelf life by 10 days. Moreover, different addition technologies may have varying inhibitory effects on microorganisms. This was demonstrated by Wang et al. [85], who found that nanoemulsions of Litsea cubeba essential oil exhibited stronger inhibition against L. monocytogenes and spoilage bacterial S. baltica compared to pure Litsea cubeba essential oil.

Q20: l.538 - necessary space before "coli";

A20: Thanks for your suggestion. In the revised manuscript, the sentence has been corrected in line 516-518 as follow:

Furthermore, lactoferrin [87], a natural antibacterial substance found in secretions like saliva, milk, and tears of mammals, can inhibit the activity of E. coli.

Q21: l.544-545 - "lactic acid bacteria" is not latin name - not italic; but...

A21: Thanks for your suggestion. In the revised manuscript, the sentence has been corrected in line 524-526 as follow:

Studies have shown that lactic acid bacteria isolated from buffalo milk curds produce bacteriocin with antibacterial properties, which can inhibit the growth of pathogenic microorganisms in chickens [89].

Q22: l.547 - "Lactococcus lactis" (uppercase "L"); and l.548 - "Gram-" (uppercase "G");

A22: Thanks for your suggestion. In the revised manuscript, the sentence has been corrected in line 526-528 as follow:

Another bacteriocin commonly used in meat is nisin, which is produced by certain strains of Lactococcus lactis and has the ability to inhibit various Gram-positive bacteria [90].

Q23: l.550 - "Lactobacillus plantarum";

A23: Thanks for your suggestion. In the revised manuscript, the sentence has been corrected in line 528-531 as follow:

Arief et al. [91] achieved significant reduction in the growth of Escherichia coli in mutton intestines by adding probiotic Lactobacillus plantarum IIA-2C12, which also preserved the color of meat products.

Q24: l.548-559 - do not use italic in "probiotic", "IIA-2C12", "garlic essential... nisin (NI)", "spoilage lactic acid bacteria";

A24: Thanks for your suggestion. In the revised manuscript, the sentence has been corrected in line 528-539 as follow:

Arief et al. [91] achieved significant reduction in the growth of Escherichia coli in mutton intestines by adding probiotic Lactobacillus plantarum IIA-2C12, which also preserved the color of meat products. However, this addition also resulted in changes to the taste of the meat and decreased acceptability to some extent. Therefore, investigating the impact of microbial preservatives on the flavor of meat products is an important area for future research. Araújo et al. [92] also investigated the bacterioreducing effects of a combination of garlic essential oil (GO), allyl isothiocyanate (AITC), and nisin (NI) on fresh intestines. The combination of 20 mg/kg NI + 125 μL/kg GO + 62.5 μL/kg AITC or 20 mg/kg NI + 62.5 μL/kg GO + 125 μL/kg AITC was found to have a significant inhibitory effect on E. coli O157H7 and spoilage lactic acid bacteria, which was better than that of single applications, without affecting the sensory acceptability of fresh intestine.

Q25: l.568 - "meat" (lowercase "m");

A25: Thanks for your suggestion. In the revised manuscript, the sentence has been corrected in line 548 as follow:

3.4. Application of hurdle technology in meat

Q26: l.588 - unnecesary space after "š";

A26: Thanks for your suggestion. In the revised manuscript, the sentence has been corrected in line 568-569 as follow:

Mikš-Krajnik et al. [6] pointed out that SAEW alone was not sufficient for complete inactivation of microorganisms.

Q27: References should be described as pointed in "Instructions for Authors". Especially, abbreviated journal name should be used. The Authors need to verify this thoroughly.

A27: Thank you for your reminder. We have read the " Instructions for Authors" and have used the abbreviated journal name as required by the journal. Please refer to the manuscript for specific modifications.
